



# Interplay of North Atlantic Freshening and Deep Convection During the Last Deglaciation Constrained by Iberian Speleothems

Laura Endres[1,2], Carlos Pérez-Mejias[3], Ruza Ivanovic[2], Lauren Gregoire[2], Anna L.C. Hughes[4],
Hai Cheng[3,5], and Heather Stoll[1]

[1]Department of Earth and Planetary Sciences, Geological Institute, ETH Zurich, Switzerland.
[2]School of Earth and Environment, University of Leeds, United Kingdom.
[3]Institute of Global Environmental Change, Xi'an Jiaotong University, China.
[4]Department of Geography, University of Manchester, United Kingdom.
[5]Faculty of Geography, Yunnan Normal University, China.

**Correspondence:** Laura Endres (endres@eaps.ethz.ch)

**Abstract.** The last deglaciation featured abrupt climate shifts driven by interactions among Earth system components, notably retreating ice sheets and meltwater input. While globally detected, the magnitude, timing, and sequence of North Atlantic source events remain uncertain. We present a Uranium-Thorium-dated stalagmite from northwestern Iberia spanning 24–12 ka BP, capturing both the impact of North Atlantic meltwater on surface ocean chemistry and regional air temperature changes. Our record reveals primarily gradual meltwater inflow during the Last Glacial Maximum and early deglaciation (about 20.8–18.2 ka BP), followed by abrupt increases during Heinrich Stadial 1. An abrupt cooling lags the first meltwater pulse by ca. 850 years, unlike later pulses. This evolving relationship between meltwater and cooling provides new constraints on the changing sensitivity of deep Atlantic convection to meltwater input throughout the deglaciation.

## 1 Introduction

The last deglaciation started in the Northern Hemisphere (NH) around 19 thousand years before present (ka BP), and marks the transition from the Last Glacial Maximum (LGM), lasting from approximately 23 to 18 ka before present (BP), into the Holocene (Clark et al., 2012). During the LGM, the NH was covered by large ice sheet complexes (Fig. 1a) covering much of North America and Eurasia (Stokes et al., 2012; Hughes et al., 2016). An increase in boreal summer insolation triggered their retreat and during their subsequent deglaciation there were episodically large amounts of meltwater (MW) and icebergs delivered into the North Atlantic (e.g., Toucanne et al. (2015); Hodell et al. (2017); Lin et al. (2021); Zhou and McManus (2024)).

It is well established, both conceptually (Denton et al., 2010; Alvarez-Solas and Ramstein, 2011; Menviel et al., 2020) and by modeling studies (Jackson et al., 2015; Klockmann et al., 2018; Romé et al., 2022), that MW contributions can cause an abrupt weakening of the Atlantic Meridional Overturning Circulation (AMOC) – the zonal integral of the surface and deep currents in the Atlantic - if the Atlantic is conditioned for instability. Such weakening, in turn, would lead to hemispheric-scale cooling (warming) in the north (south), and trigger atmospheric $CO_2$ rise (Broecker et al., 2010). While there is evidence for





several episodes of AMOC weakening and NH cooling during the deglaciation (Mcmanus et al., 2004; Shakun et al., 2012; Ng et al., 2018), linking these to specific melt discharge events remains challenging. Uncertainties in ice geometry, even at the scale of total volume and global distribution (Stokes et al., 2015), propagate into the derived meltwater histories mostly used for transient deglacial simulations (Bethke et al., 2012; Sherriff-Tadano et al., 2018; Kapsch et al., 2022; Snoll et al., 2024). Furthermore, direct evidence for meltwater from palaeoclimatic records is sparse and retrieving accurate radiocarbon ages from marine cores on such short centennial time scales is hindered by the increased stratification during weak AMOC states (Peck et al., 2006; Peck, 2017). Generally, time scales relevant for melt events range from a few years to a few centuries, a time range in which the North Atlantic ocean and its adjoint ice sheets are not able to equilibrate (Wunsch and Heimbach (2008), Fig. 1c), emphasizing the need for more regional records to reconstruct their sequence and impact on the Earth system. Even more so, as modulations of AMOC strength depend on background climate (Zhu et al., 2015), and can occur directly forced by freshwater (Zhu et al., 2017; Snoll et al., 2024), spontaneously (Romé et al., 2022; Armstrong et al., 2023) or delayed (Alvarez-Solas and Ramstein, 2011; Romé et al., 2024). This adds an additional layer of complexity to the synchronization of regional melt events with global records.

To move forward our understanding of the relationship between melt-induced freshening and AMOC strength, we are building upon a direct freshening mechanism by detecting the $^{18}$O-depleted MW in speleothem $\delta^{18}$O proximal to the melt water source regions, an effect previously documented in the Northwest Iberian Speleothem Archive (NISA) (Stoll et al., 2022) and model simulations (Zhu et al., 2017). Here, we provide a new decadally-resolved isotopic record from NISA, 24-12 ka BP, to constrain the timing of MW addition and elucidate the relationship between melt-induced freshening, North Atlantic temperature changes and AMOC strength throughout the entire last deglaciation. Given the coastal location of our site and prevailing atmospheric conditions induced by Earth's rotation, the direct effect of $\delta^{18}$O-depleted MW on the surface ocean is captured by $\delta^{18}$O in the speleothem, allowing us to infer the history of NH MW at high temporal resolution. In addition, we provide a coeval record of relative temperature from speleothem $\delta^{13}$C, which we have corrected for in-cave fractionation effects ($\delta^{13}$C$_{\text{init}}$). Since both stable isotopes are measured on the exact same samples, this allows a direct and high-resolution study of the temporal relationship between NH MW ocean in-flux and temperature change in the North Atlantic realm.

## 2 Material and Methods

### 2.1 Cave site, sample, preparation, and sampling procedure

Modern precipitation at the cave site is dominated by cool season rainfall, but many drips remain active also during the summer period of lower precipitation-evapotranspiration. Speleothem Glas has been selected for this detailed study because of the absence of any signs for non-continuous growth and for featuring a homogenous trace element structure, also when diverging from the main growth axis (SI Appendix, Fig S1-S2). We focus on the 4.2 cm interval of Glas bracketed by a growth discontinuity on the older part and significant decrease in growth rate on the younger boundary (Fig. 2). Due to the slow growth rate and the presence of abrupt climate changes over the early last deglaciation, the sampling strategy for U/Th dating has been optimised by continuously micro-milling the sample with a 50 μm trench along growth layers, measure stable isotopes on






## 2.2 U-Th dating and age model

Despite small sample sizes (20-40 mg), good analytical errors of less than 100 years have been achieved in each sample. U-
Th samples were analyzed with a ThermoFisher Neptune plus Multicollector Inductively Coupled Plasma Mass Spectrometer
(MC-ICP-MS) at Xi'an Jiaotong University, following the methodology and decay constants reported in Cheng et al. (2013),
see Tab. S1. The age model used in this study (Fig. 2) has been computed with BChron (Haslett and Parnell, 2008), and has
been optimised for highest correlation with Sr/Mg ratio.


## 2.3 Stable isotope and trace element ratios measurements

Stable isotope ratios were measured at ETH Zurich with a Thermo-Finnegan Delta V Plus coupled to Gas Bench II (Breitenbach
et al., 2015). Carbonate powder for trace element analysis were dissolved in $2\%$ $HNO_3$ and analyzed for trace element to
calcium ratios employing an Agilent 8800 QQQ ICP-MS at ETH.


## 2.4 Correction for in-cave fractionation

Prior calcite precipitation (PCP) occurs when parts of the initial dripwater solution already supersaturate and precipitate due
to the exchange with cave air before reaching the stalagmite. PCP can alter the $\delta^{13}C$ signature significantly, by preferentially
removing $^{12}C$ from the remaining dripwater solution (Rudzka et al., 2011; Fohlmeister et al., 2020). To focus on the temperature
signal preserved in soil $CO_2$ and consequently $\delta^{13}C$, the record was corrected for PCP by using the Mg/Ca ratio as a guide
line for the amount of degassing occurring, following the methodology of Stoll et al. (2023), with details being described in
SI appendix. In contrast, in most instances, the actual bedrock $\delta^{13}C$ is of minor relevance, as the system can be assumed to be
rather open and therefore the $CO_2$ in the soil overwhelms the rock signature (Lechleitner et al., 2021).


## 2.5 Breakpoint determination

We constructed a piecewise linear model to determine the breakpoints using a smoothed 200-year rolling mean time series with
the R package 'segmented'. The algorithm uses taylor-expansion to iteratively approach the best estimate, starting from initial
breakpoint guesses and providing standard errors for each breakpoint (Muggeo, 2003). For this study, the plotted uncertainty
is given as the integrated $2\sigma$-uncertainty of the breakpoint estimation and the age model ensemble.




## 2.6 Climate model experiments and NISA Melt Source Contribution Index

We reemploy an already published HadCM3 model simulation (Romé et al., 2022) to examine the progression of a freshwater anomaly from different source origins and under different AMOC states to the NISA location, the model is described in the Appendix. The original simulation includes 10,000 model years and captures a glacial climate state with an AMOC that oscillates between a strong (relatively 'warm' climate) and a weak (relatively 'cold' climate) state, with about 1,500 year periodicity, triggered by a constant meltwater flux corresponding to a reconstructed ice sheet history at 17.8 ka BP (GLAC-
1D, Ivanovic et al. (2016)). The seasonal averages of surface air temperature and precipitation - evaporation, plotted in Fig. 1c-d, were computed within the region confined by an red border in Fig. 1b. To distinguish between a strong and a weak state AMOC within the simulation, we followed the methodology of Romé et al. (2025), where the years that fulfilled each condition were selected based on changes in the mixed layer depth (MLD) in key convection sites (Irminger and GIN Seas). In these simulations, Irminger Sea MLD is in the range 17-25 m during the weak AMOC state, deepening to 25-65 m during the strong
AMOC state. GIN seas MLD is 27-40 m and 65-105 m, respectively. From the full simulation, 2,367 years fulfilled the 'weak' condition (average AMOC strength at 26.5°N: 8 Sv), and 1,488 years fulfilled the 'strong' condition (16 Sv), coined 'zonal' in Romé et al. (2022). Segments of the simulation were re-run for our study, adding a set of passive ocean tracers tagging the different likely meltwater source regions to assess the potential of each region to influence the isotope signal at our study site. The results are shown as a qualitative NISA Melt Source Sensitivity Index (Fig. 1e). To calculate the index we use the estimated
moisture uptake at our site from a hysplit analysis based on rainfall in NISA location El Pindal from 2015-2016 (González, 2019) to scale the uptake from the model's surface ocean dye anomalies (Fig. 1b). The index, then calculated as a percentage of the mean total arriving meltwater at the study site during a strong AMOC state after 200-300 years of continuous injection, gives a relative estimate of the importance and the variability of the signal in the surface ocean.

## 3 Results and Discussion

### 3.1 Sensitivity of NISA to surface ocean freshening and cooling

At NISA cave locations (Fig. 1), precipitation is mainly controlled by westerly winds and the passage of Atlantic fronts while the Cantabrian mountain ridge reduces the influence of water vapor transport from the south (Moreno et al., 2021), making rainfall from the proximal North Atlantic the dominant moisture source. Over the penultimate and the last deglaciation, speleothem $\delta^{18}O_{NISA}$ closely tracks the long term trend in $\delta^{18}O$ of seawater ($\delta^{18}O_{sw}$) reconstructed from marine archives
from the eastern North Atlantic (Stoll et al., 2022). This strong response to the moisture source composition occurs because in this setting, the temperature-dependent fractionation of $\delta^{18}O$ into calcite is of a similar magnitude but opposite sign to the temperature dependence of $\delta^{18}O$ in precipitation (McDermott et al., 2006; Stoll et al., 2015), which mitigates additional speleothem $\delta^{18}O$ variability due to temperature changes. Additionally, due to the coastal cave location, effects on $\delta^{18}O$ due to changes in altitude or prior rainout are minimized.





**Figure 1.** Overview of sites referenced in this study, the expected meltwater signal originating from different ocean bassins, and the expected environmental changes related to an AMOC decline. **(a.)** A Map including the study site NISA (red-rimmed white star) and other sediment cores used in this study for proxy comparison (red circles), namely MD95-2002 (Toucanne et al., 2015), U1308 (Hodell et al., 2017), Hu97048-07 (Rashid et al., 2012), and MD02–2550 (Wickert et al., 2023). The maximum LGM ice sheet extent (Batchelor et al., 2019) (grey, blue border) is illustrated for the main northern ice sheet sectors: Laurentide (LIS), Greenland (GIS) and the European Ice sheet (EIS) sectors: Svalbard-Barents-Kara (SBKIS), Scandinavian (SIS) and British-Isles (BIIS). Ocean colouring illustrates the simulated mean annual sea surface salinity anomaly expected for Heinrich Stadial 1 (Ivanovic et al., 2018) due to meltwater input and associated AMOC decline. **(b.)** A Map showing the oceanic moisture source areas from a Hysplit analysis on precipitation monitoring data at NISA cave sites (red-rimmed white star) from 2015-2016 (González, 2019). The colored regions denote three composite Northern hemisphere ice sheet's meltwater source regions: the GIN Seas and northeastern North Atlantic (purple), the Arctic Ocean (petrol), and the Gulf of Mexico and Labrador Sea (orange). These regions were further used to compute the regional NISA Melt Source Sensitivity Indexes in (e.). **(c.,d.)** Based on a simulation simulating regional meltwater discharge at 17.8 ka, re-employed from Romé et al. (2022), the expected difference at our NISA study site (averaged over the red rectangular region) between different AMOC states for **(c.)** surface air temperature and **(d.)** moisture availability (precipitation - evapotranspiration; P-ET). **(e.)** Barplot shows the NISA Melt Source Sensitivity Index (SI Appendix, Methods); a relative mean contribution estimate of how much meltwater reaches the NISA location from each of the three regions after 200-300 years of continuous injection during a weak (blue) and strong (red) AMOC state, based on a simulation using meltwater input at 17.8 ka BP (Romé et al., 2022).



On a centennial time scale, a freshwater anomaly originating from a certain ice sheet sector will not spread evenly across latitudes and depths. This may amplify or attenuate the impact of a given meltwater flux on the proxy archive. To test the dependency of source origin and AMOC background state, the distribution of meltwater from different NH ice sheet sectors has been tracked by adding conservative dye tracers to the simulations of Romé et al. (2022). We compute the relative intensity of the tracer in the proximal sector of the modern NISA moisture source region (Fig. 1b). The sensitivity of the tracer to the

source region and AMOC state is expressed as the NISA Melt Source Sensitivity Index (Methods, SI Appendix). The Melt Source Sensitivity Index demonstrates that for a given meltwater flux (Fig. 1b,e), the signal from ice sheet meltwater reaching the NISA sites is higher when AMOC is in a weak state than in a strong state, regardless of the melt source region. This is because a stronger AMOC more rapidly redistributes meltwater throughout the global ocean. Furthermore, we can identify that under the weak AMOC state, the tracked NISA source region more strongly accumulates meltwater delivered to the GIN seas

(of Eurasian origin), whereas meltwater discharged by the Laurentide ice sheet to the western North Atlantic more strongly accumulates in the tracked region under strong AMOC conditions (Fig.1b). For both, weak and strong AMOC, meltwater anomalies originating from the Arctic will already largely propagate to the subsurface waters and have limited impact on the surface North Atlantic $\delta^{18}O_{sw}$.

    Northwestern Iberia is also particularly sensitive to AMOC-driven changes in temperature (global maps in SI Appendix,

Fig. S3). A switch from a strong to weak AMOC state in the early deglaciation drives surface air temperature changes of up to -5° C in NW Iberia for all seasons in the same simulations (Fig. 1c). In contrast, the moisture availability within the region remains similar across such transitions (Fig. 1d). In our record, $\delta^{13}C_{init}$ provides a qualitative measure of regional surface air temperature. This is because the $\delta^{13}C$ of cave dripwater is initially set by the exchange of infiltrating water with $CO_2$ in the soil and epikarst. In moisture-replete regions such as NW Iberia, warmer temperatures stimulate higher heterotrophic and

autotrophic respiration rates and raise soil $CO_2$ (Romero-Mujalli et al., 2019). Because respired $CO_2$ is much lower in $\delta^{13}C$ than atmospheric $CO_2$, warm periods of high soil $CO_2$ lead to a more negative $\delta^{13}C$ signature in cave dripwater. While in-cave processes of degassing and prior calcite precipitation (PCP) can in some cases subsequently modify the $\delta^{13}C$ signature in cave dripwater (Mühlinghaus et al., 2007; Mickler et al., 2019), recent approaches using Mg/Ca as an independent PCP indicator allow calculation of the initial $\delta^{13}C_{init}$ defined by the soil and vegetation processes from the measured $\delta^{13}C$ (Stoll et al., 2023;

Lechleitner et al., 2021). We thus apply this technique to derive the temperature sensitive $\delta^{13}C_{init}$ (Methods, SI Appendix).

### 3.2   The deglacial North Atlantic climate record of Glas

    The stalagmite named 'Glas' originates from the NISA cave 'La Vallina' (4.8067° W, 43.4100° N), located 4 km from the modern coastline. Twenty-five new U/Th dates, with analytical errors of less than 100 years, constrain the chronology of Glas between 24 and 12 ka; and potential variations in growth rates are assessed through comparison with Sr/Mg measurements

(Fig. 2, Methods, SI Appendix, Tab. S1, Fig. S1). Annual growth rate averages 4 μm, but drops to 1 μm around 13 ka. By using combined aliquots of the same sample powder for Uranium-Thorium dating as have been used for high-resolution stable isotope analysis, a fully consistent age model with minimized depth uncertainties is attained for the sequence of isotopic events



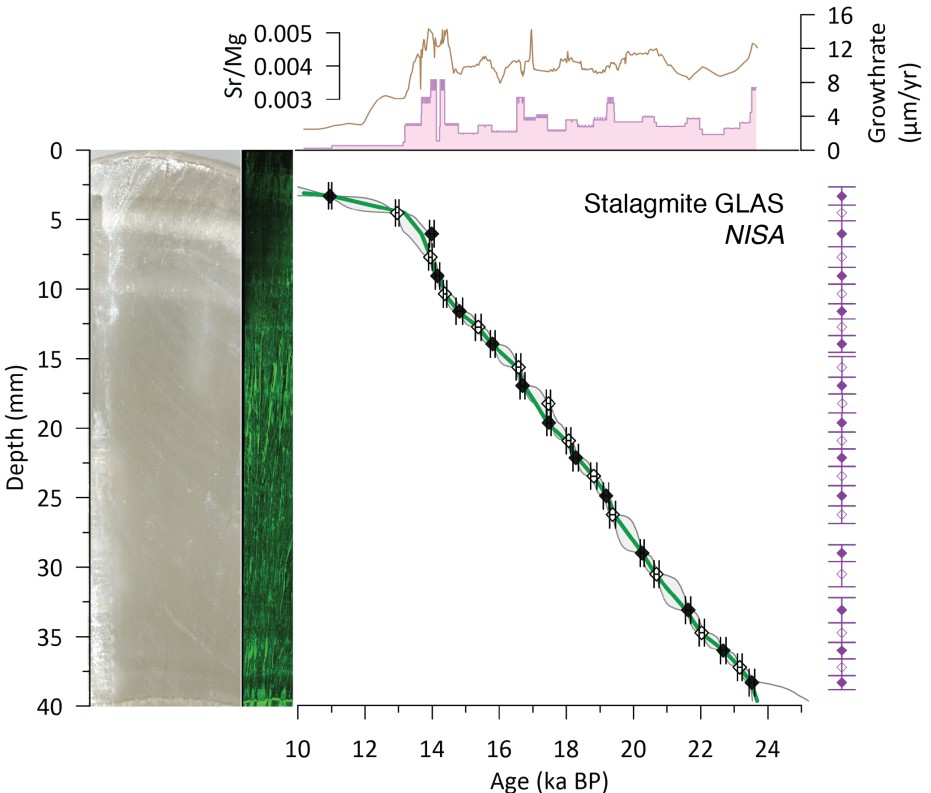

**Figure 2. Age model of NISA Stalagmite Glas.** Main panel from left to right: scan of the slab used for analysis (grey shading); the corresponding Confocal laser scanning microscopy image (green shading); and the constructed BChron age model (green line) used in all further analysis (Methods), based on the U-Th ages (yellow dots, bounded by horizontal errorbars denoting the $2\sigma$ uncertainty); and, on the right, actual depth of the combined, original samples used in this study (purple vertical errorbar). Note that our sampling procedure ensures no shift between the assigned depths for dating and isotope track. In the top panel, growth rate is calculated based on the BChron age model (pink fill), and we use Sr/Mg ratio (brown line) as an additional indicator for changes in growth rate to confirm the age model from the U/Th dating sequence.

despite the slow growth rate. Featuring this improved age model, Glas supersedes the prior NISA stack for the last deglaciation, but the main features of Glas have been validated in other NISA records (SI Appendix, Fig. S2).

## 3.3 North Atlantic surface ocean freshening history

### 3.3.1 LGM evolution

Our new record from Glas captures the end of Heinrich Stadial 2 (HS2) with a trend to more saline conditions in the North Atlantic indicated by the shift to a heavier $\delta^{18}O$ signature and the coincident rapid recovery to warmer temperatures evident by a shift to more negative $\delta^{13}C_{init}$ (Fig. 3). The observed warming likely reflects a strengthening of AMOC after HS2, which





results in a more efficient dispersal/removal of $\delta^{18}O$ anomalies from the surface ocean and is in accordance with the previously discussed NISA melt source sensitivity index (Fig.1e). Our chronology is consistent with the proposition that the HS2 phase of Southern BIIS retreat and associated enhanced MW flux to the ocean ended around 23.5 ka, aligning with the Southern European Ice sheet (EIS) events found in sediment cores (Toucanne et al., 2015). The ice sheet model GLAC-1D (Ivanovic et al., 2016), which has been used to force transient deglacial climate simulations in a number of studies previously (Snoll et al.,

2024), suggests a slightly later timing of peak meltwater delivery from the Southern Laurentide ice sheet (LIS) via the Labrador Sea+GoM outlets as well as some meltwater discharging into the Arctic region (Fig. 3a-b.; extended discussion and ice sheet mass loss maps in SI Appendix, Fig. S5). Thus, the new Glas record suggests that either the GLAC-1D model shows these elevated melt events around 500 years too late, or despite a persistently high meltwater discharge, AMOC has restrengthened and successfully dissipated the surface $\delta^{18}O$ anomaly from a LIS meltwater pulse postdating HS2.

A subsequent episode of melting from 22.5 to 21.5 ka is suggested by a small, gradual decrease in $\delta^{18}O$ of -0.3 ‰, which returns to previous levels between 21.5 and 20.5 ka. Sustained freshening begins around 20.5 ka, marked by a gradual decrease of $-0.5‰$ $\delta^{18}O$ over 2 ky (Fig 3). This progressive freshening confirms the presence of MW in the eastern North Atlantic from the Scandinavian ice-sheet (SIS) as previously suggested by Nd isotope provenance of detrital minerals delivered into the Bay of Biscay (Toucanne et al., 2015).

### 3.3.2 Abrupt freshwater onset at 18.04 ka

An abrupt freshening (-0.58 ‰$\delta^{18}O$ in $382^{+164}_{-94}$ years) starts at $18.04^{+0.16}_{-0.09}$ ka BP, which we interpret as the onset of Heinrich stadial 1; HS1a in Fig 3, with other regional speleothem records also indicating negative $\delta^{18}O$ shift around this time (SI Appendix). This freshening may reflect accelerated ice sheet melting in various regions (Fig. 3c). For example, there is evidence of elevated freshwater drainage from the Southern Laurentide to the Gulf of Mexico (Wickert et al., 2023), and reconstructions of EIS evolution (Hughes et al., 2016; Clark et al., 2022) suggest a strong likelihood that SIS and BIIS separated by 18 ka

BP. The river-routed GLAC-1D ice model time series suggests discharge mainly into the Arctic, but since our dye tracking indicates that freshwater discharged into the Arctic produces only a weak signal in our speleothem record (Fig. 1e), the strong freshening signal recorded by Glas at 18.04 ka BP rather indicates that at least part of the melt from Svalbard-Barents-Kara ice sheet (SBKIS) retreat was delivered to the GIN Seas and/or down the Channel River. This route would be consistent with the high terrestial input through the channel river recorded at that time (Toucanne et al., 2015).

The phase of elevated freshening is followed by a period of sustained low $\delta^{18}O$ (here HS1b), indicating that the sources and sinks of a MW-induced $\delta^{18}O$ anomaly were in balance over the next 1500 years. A negative $\delta^{18}O$ value from a freshwater pulse can be subsequently sustained despite lowered meltwater delivery if AMOC is weak, especially if the MW discharge region (i.e. GIN seas) is close to our study site (Fig. 1b). In this case, sustained $\delta^{18}O$ values in a weak AMOC state could potentially

also mask limited ice sheet regrowth, as suggested in Wickert et al. (2023). Alternatively, marine records from the Hudson Strait indicate elevated meltwater delivery to the Atlantic (Rashid et al., 2012; Denton et al., 2021), which could provide an additional source to sustain the Glas $\delta^{18}O$ anomaly provided that the AMOC remains relatively vigorous.





### 3.3.3 Transient freshening from 16.5 ka through the Bølling-Allerød

A first smaller transient event with onset at $16.5^{+0.09}_{-0.31}$ ka BP and a change of - 0.5‰ $\delta^{18}$O, is followed by two larger events
at $16.22^{+0.24}_{-0.13}$ (HS1c in Fig.3e) and $15.44^{+0.19}_{-0.25}$ ka BP with a shift around -0.79‰ $\delta^{18}O$ each. All events are short-lived and dissipate from the subtropical North Atlantic in a few hundred years. The latter two events coincide with the radiocarbon age estimates for two distinct central North Atlantic ice rafted debris (IRD) peaks (Fig. 3c), which have been interpreted as two episodes of enhanced iceberg calving from the Laurentide Ice Sheet related to Heinrich Event 1 (Hodell et al., 2017). Such a twin-peak structure has also been achieved with simulations considering dynamic meltwater re-routing, which produces
a second wave of enhanced Laurentide iceberg calving around 1,000 years after the first episode (Ziemen et al., 2019). In addition, all events feature an abrupt onset of the negative $\delta^{18}$O anomaly but a gradual dissipation of the anomaly, which is consistent with a binge-purge mechanism suggested for Heinrich events (MacAyeal, 1993). Alternatively, the timing of the later $15.44^{+0.19}_{-0.25}$ ka freshwater pulse is within age uncertainty of the most likely timing for Cordillerean-Laurentide (CIS-LIS) ice saddle collapse ($15.226_{-0.81}^{+1}$ ka; Norris et al. (2022)).
By 14.5 ka, the $\delta^{18}$O had relaxed back to values characterizing the plateau of HS1b. However, the remainder of the Glas record (until 12.5 ka) exhibits higher variability in $\delta^{18}$O, with numerous high amplitude, short lived freshening pulses. We propose that this ice sheet shrinkage causes a greater vulnerability to climate change, creating more reactive ice sheets, and that together with a stronger AMOC (compared to the earlier deglaciation) this explains the observed higher variability in $\delta^{18}$O. Surprisingly, the rapid, large (14-18 m) sea level rise termed meltwater pulse 1A, thought to occur slightly after the
start of the BA around 14.6 ka (Deschamps et al., 2012; Gregoire et al., 2016), is not manifest as a salient freshening anomaly in our record. Potentially, the amplitude of Meltwater pulse 1A is lower than previously described, has a larger contribution from the southern hemisphere and/or part of this sea level rise is reflected in the meltwater pulse we detected at 15.44 ka BP. Alternatively, a more vigorous AMOC may have efficiently downwelled the surface freshwater anomalies, so that freshening is not detected in the surface North Atlantic moisture source area of Glas.

## 3.4 Iberian temperatures and relationship to freshening

### 3.4.1 AMOC insensitive to gradual freshening during the LGM

The new $\delta^{13}$C$_{\text{init}}$ record from Glas provides a direct speleothem chronology for a relative regional temperature evolution in the North Atlantic region (Fig. 3f). The Glas record highlights the warmer temperatures of the early LGM, followed by a significant cooling 21.5-20.5 ka BP to values sustained through the late LGM. This cooling is likely a global cooling, as it follows closely
the timing of the LGM sea level lowstand (Yokoyama et al., 2018), whereas the AMOC does not show significant variations over this time based on Pa/Th ratio Fig.3g). During the two periods of very gradual freshening of the early and late LGM intervals, the stable temperatures indicated by $\delta^{13}$C$_{\text{init}}$ suggests that AMOC is too stable and/or the MW forcing is too gradual and insufficient to disrupt convection at this time.





**Figure 3.** Speleothem record Glas and its relationship to last deglaciation ice sheet history and AMOC strength. **(a)** Snapshots of GLAC-1D ice sheet history at key intervals (Ivanovic et al., 2016). **(b)** River-routed MW delivered to the three key ocean regions identified in Fig. 1, computed in (Romé et al., 2022) from the GLAC-1D ice sheet reconstruction. **(c)** Simplified history of proposed MW discharge events from different Northern Hemisphere ice sheet sectors in sedimentary archives, broadly: the Northern Laurentide (Denton et al., 2021; Rashid et al., 2012), Southern Laurentide (Wickert et al., 2023), Channel River discharge (Toucanne et al., 2015), and distally transported iceberg rafted debris (IRD) concentration at site U1308 (Hodell et al., 2017); simplified (SI Appendix, Fig. S4). The new data of Glas (this study) is plotted in **(d)** the position of U/Th ages and their associated analytical age uncertainty, **(e)** $\delta^{18}O$, used as a proxy for surface ocean freshening, and **(f)** $\delta^{13}C_{init}$, used as a proxy for surface air temperature above the cave. **(g)** 9-point average composite Pa/Th ratio as a qualitative AMOC strength indicator (Ng et al., 2018). The brown fill reflects the 2-$\sigma$ standard error of the composite and the grey dots the individual composite data points.



### 3.4.2 Delayed AMOC response to HS1 freshening

The rather sustained temperatures over the late LGM in northern Iberia are followed by a rapid decline to coldest temperatures during the earliest part of HS1. This cooling falls within the broad period of decreased AMOC strength diagnosed from Pa/Th composite ratio; Fig.3g), with the Glas record featuring a greater chronological precision and resolution. During the early stages of the last deglaciation, temperature changes are typically not manifest in Greenland ice core (NGRIP) $\delta^{18}O$ because a decrease in winter rainfall changes the overall annual $\delta^{18}O$ signature and cancels the temperature effect (He et al., 2021b). Critically,

compared to the transient freshening starting at 18.04 ka BP, the $\delta^{13}C_{init}$ indicates that regional surface air temperature initially remained stable, but then significantly cooled only around 850 years later, with an onset at $17.17^{+0.34}_{-0.12}$ ka BP. Because both isotope records derive from the same powders, this reflects a genuine lag in response time. As has been shown by model studies (Romé et al., 2022; Van Westen et al., 2024), the abrupt temperature change within the European realm and at NISA cave locations is a key characteristic of a transition into a weak AMOC state (see Fig. 1c). We, accordingly, interpret the Glas

record such that, even though subtropical North Atlantic surface freshening peaked at 18.04 ka BP, most effective AMOC weakening did occur around 17.17 ka BP and the coldest temperatures at our site for the last deglaciation were reached only by $16.87^{+0.31}_{-0.16}$ ka BP. Further, while the composite Pa/Th record is smoothed and points towards less abrupt decrease in AMOC strength, individual cores contributing to the composite rather feature more abrupt changes as well (Ng et al., 2018).

Although in many cases the weakening of AMOC scales directly and almost immediately with freshwater forcing (Snoll

et al., 2024), this is not necessarily true if the forcing is not strong enough, not well located, or the background state of the ocean is not (yet) sensitive enough (Van Westen et al., 2024; Romé et al., 2025). We propose that delayed cooling after the freshwater input with onset at 18.04 ka reflects the need to build up a positive subsurface salinity anomaly or/and a required $CO_2$ level to cross an AMOC threshold and trigger its weakening. The required delay is expected to depend on the meltwater forcing properties, internal ocean structure and ice sheet geometry (Kapsch et al., 2022; Romé et al., 2025). Additionally,

already before the rapid decline, $\delta^{13}C_{init}$ shows early signs of a prior slight weakening, which would be consistent with a mechanism involving the build-up of subsurface anomalies (Alvarez-Solas and Ramstein, 2011). The period of most extreme cooling was brief and recovered back to near late-LGM levels without any evident change in North Atlantic surface freshwater conditions.

The sequence of the Glas record and its comparison to radiocarbon dated IRD events (Hodell et al., 2017) both suggest that

cooling has preceeded the large discharge of icebergs into the North Atlantic associated with Heinrich Event 1 by a bit more than 800 years. A similar sequence of events with a lag between initial surface ocean cooling by planktic Mg/Ca and peak IRD abundance was recorded within a single core on the Iberian Margin (Skinner and Shackleton, 2006).

### 3.4.3 Temperature response to transient freshwater from 16.5 ka onward

Coinciding with the midpoint in the shift in $\delta^{18}O$, at $16.14^{+0.28}_{-0.25}$ ka BP (HS1c in Fig 3e), is a very short-lived cooling, for which

the precise duration and magnitude cannot be well constrained due to very slow growth rates. Notably, a similarly short-lived cooling also accompanied the largest freshwater pulse of the Penultimate Deglaciation (MWPTII-B around 134.25 ka BP, (Stoll





et al., 2022)). Such brief episodes could link to the presence of faster, potentially atmospheric, pathways, reorganizing global circulation patterns (Markle et al., 2017; Fohlmeister et al., 2023). In sharp contrast, no temperature response is present for the freshwater pulse at 15.44 ka (HS1d), which we hypothesize indicates that this pulse was not able to alter AMOC strength any
further.

### 3.4.4    Role of MW in the Bølling-Allerød

The progressive decay of the MW anomaly in HS1d between 15 and 14.7 ka suggests that the anomaly in the eastern North Atlantic was diminishing already before the abrupt onset of the Bølling-Allerød (BA) warming, which is recorded in Glas at 14.77-14.47 ka BP, a timing indistinguishable from that in NGRIP (North Greenland Ice Core Project members, 2004)) (Fig
4). The abrupt BA warming coincided with a subsequent steep reduction in freshwater in the North Atlantic, which may reflect accelerated advection and downwelling due to AMOC strengthening (Fig. 1b). While having less freshwater in the North Atlantic may have facilitated the resumption of AMOC at the BA onset, the Glas record suggests a surface freshwater anomaly comparable to the end of the prior freshening event (HS1c), where there was no evidence of AMOC restrengthening. Thus, instead of NH MW, the initial trigger for BA warming might be the crossing of a AMOC sensitivity threshold related to the
evolving climate and ocean boundary conditions or located within the Southern Hemisphere cryosphere.

Within BA, the $\delta^{13}C_{init}$ record indicates two warm phases separated by a cooling initiating around 14ka, potentially consistent with the onset of the Older Dryas (e.g., as seen in NGRIP (North Greenland Ice Core Project members, 2004)) . However, the slowed growth rate in Glas complicates resolution of the precise chronology of this cooling event. Nevertheless, during the early BA temperature seems to follow the changes in $\delta^{18}O$ closely, which would be consistent with AMOC strength following
MW input such as in most hosing simulations (Snoll et al., 2024), whereas no clear impact on temperature is present during the later BA, suggesting a change towards lower sensitivity to MW inputs.

### 3.5    Impacts beyond the North Atlantic Realm

We evaluate the relationship of North Atlantic abrupt freshening and temperature change events to other aspects of the global climate system recorded in archives posessing an absolute layer counted ice core or speleothem U-Th chronology (extended
discussion in SI Appendix, Fig. S6). We focus on the comparisons of four prominent change points in Glas (Methods, Fig. 4a): the $\delta^{18}O$-indicated freshening events E1 (E1 freshening: 18.04 - 17.65 ka BP, E1 temperature: 17.18 -16.88 ka BP), E2 (16.22-15.99 ka BP) and E3 (15.44-15.19 ka BP) and the $\delta^{13}C$ -indicated abrupt warming event E4 (14.77 - 14.47 ka BP).

### 3.5.1    Monsoon and the Westerly Jet over Asia

Isotope-enabled models suggest that millennial scale positive anomalies in the $\delta^{18}O$ in rainfall and speleothems from the East
Asian summer monsoon domain (EASM) reflect reduced intensity of convection and rainfall over the Indian summer monsoon and southward-shifted westerly jet enhancing southern EASM rainfall and reducing northern EASM rainfall (He et al., 2021a). Across the 18.04 ka event (E1f), in the EASM record from Hulu Cave (Fig 4d), a positive shift in $\delta^{18}O$ suggests a southward



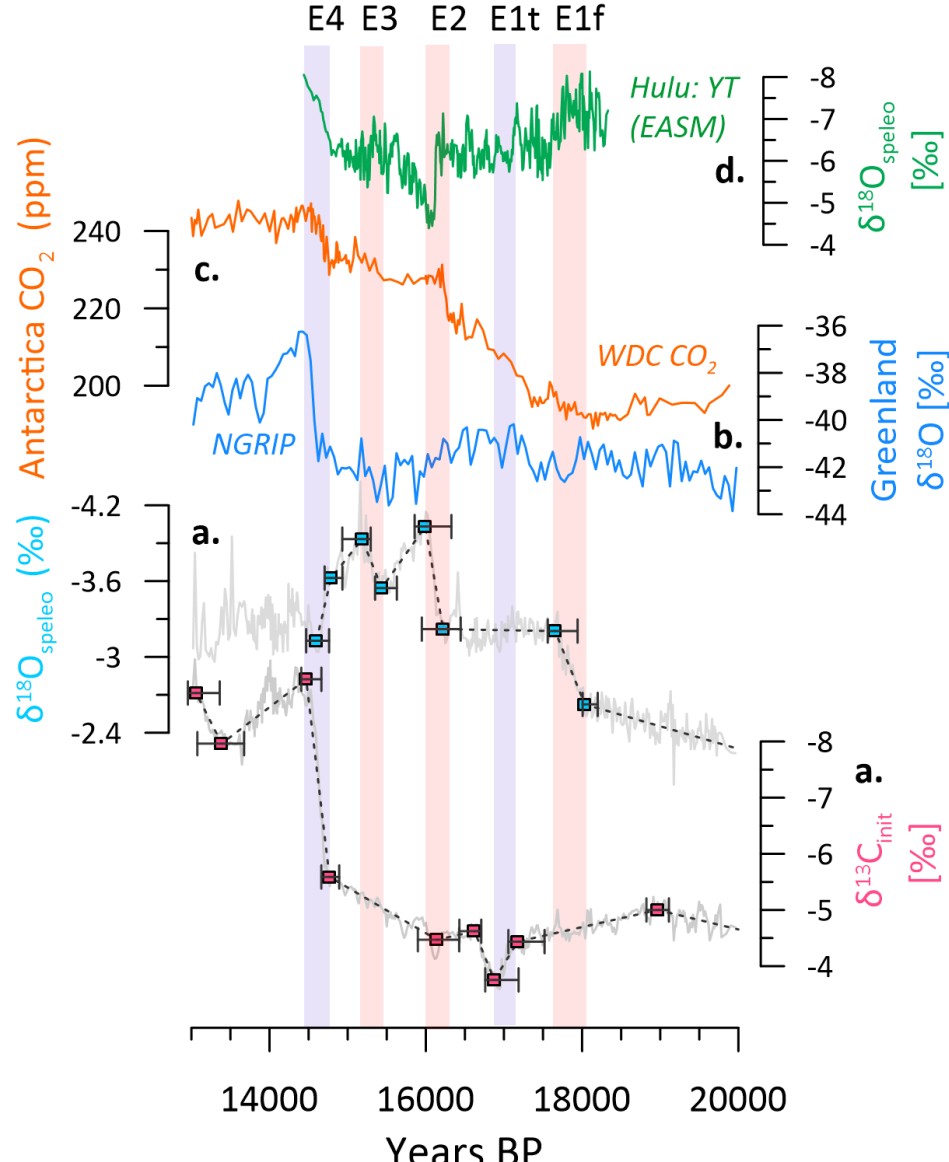

**Figure 4.** Abrupt events over the last deglaciation recorded in Glas and their temporal relationship to published proxy data. Glas (**a.**, this study) is compared to **(b.)** the Greenland ice core on GCC05 timescale (North Greenland Ice Core Project members, 2004), **(c.)** $CO_2$ from West Antarctica ice core (Marcott et al., 2014) and, **(d.)**, Chinese Speleothem record YT from Hulu Cave recording changes in the East Asian Summer Monsoon (EASM) (Zhang et al., 2014). Red shadings mark three identified transient freshening events (E1-E3), blue shading the temperature events (E1t,E4). Blue and pink square symbols in (a.) mark the change points in Glas $\delta^{18}O$ and $\delta^{13}C_{init}$ found through segmented linear regression. The errorbars denote the combined $2\sigma$-uncertainty from the full BChron age model ensemble and the change point detection.





shift of the westerly jet concurrent or slightly lagging the North Atlantic freshening. Thus, the shift in the westerly jet may have preceeded the most abrupt AMOC reduction (E1t), but could instead be related to the large changes European ice sheet

sectors, namely changes in SBKIS and the separation of SIS and BIIS, affecting the atmospheric circulation already prior to the extreme AMOC weakening. The following transient freshening at 16.22 ka (E2) is manifest in a concurrent abrupt positive $\delta^{18}O$ shift indicating a rapid southward shift of the Westerly jet in phase with the North Atlantic freshening. The recovery of the jet position is coincident with the relaxation of the freshwater anomaly. The 15.44 ka MW pulse (E3), in contrast, has no corresponding changepoint in the EASM record. This may reflect that it was another pulse of freshwater into an already

weak AMOC, which triggered no further AMOC weakening or further associated shifts in atmospheric patterns. Such an interpretation also strengthens our prior hypothesis that the surface freshening anomaly originates from a source that makes it well visible in Glas, but less effective in further disrupting AMOC, such as meltwater from the Gulf of Mexico (Fig. 1b), consistent with the most likely timing of CIS-LIS breakup (Norris et al., 2022). In addition, any further effect of freshwater on the Westerly Jet may have been compensated for by the atmospheric effects of changes in LIS height (Ullman et al., 2014). At

the onset of the BA event, the rapid southward shift of the Westerly Jet is synchronous with the abrupt warming, confirming previous interpretations of a coherent global response to AMOC reinvigoration (Pöppelmeier et al., 2023). Overall, our findings imply that while millennial scale variations in the $\delta^{18}O$ from speleothems in the EASM do share many features with the North Atlantic realm, it is not an exact copy of its ocean dynamics and caution is needed when tuning the centennial to millennial scale features of North Atlantic records directly to an EASM signal.

### 300 3.5.2 Links between AMOC and atmospheric CO$_2$ rise

The onset of the first rapid deglacial atmospheric rise in CO$_2$ at 18.1 ka BP (Fig. 4b) has been hypothesized to be "broadly coincident" with the reduction in AMOC strength (Marcott et al., 2014), but hitherto, this has been difficult to evaluate since the NGRIP ice core does not mark well the HS1 cooling. The results from Glas indicate that the first centuries of rapid CO$_2$ rise, like the shift in the EASM, began with, or slightly after the freshening but before the onset of abrupt cooling. This suggests that

the first centennial carbon cycle response is linked to rapid atmospheric reorganization which may predate the main AMOC weakening. Thereafter, the millennial scale CO$_2$ increase corresponds to the coldest period in NW Iberia, interpreted to reflect the weakest AMOC (Fig 4c). Around 16.2 ka, the abrupt centennial CO$_2$ rise is coincident with the abrupt freshening (E2) and southward shift of the Westerly jet manifest in the EASM, suggesting an abrupt synchronous atmospheric reorganization. Our data suggest a brief cooling at the same time, which could be an expression of a very brief abrupt AMOC weakening or

also North Atlantic sea ice expansion in response to freshening. After this step, there is a long plateau of CO$_2$ concentration, but despite the lack of a AMOC response in Glas, there is a small CO$_2$ excursion at the freshening event of 15.44 ka (E3). It has been suggested that the plateau may reflect a balance between ocean CO$_2$ emission and uptake of CO$_2$ by regrowth of the terrestrial biosphere (Marcott et al., 2014), which may be consistent with continued reduction of the NH ice sheet area evidenced by persistent MW pulses. Alternatively, the CO$_2$ content of waters ventilating in the Southern Ocean may be

stabilizing through this later part of HS1 (Yu et al., 2022).



## 4 Conclusions

The new Northern Iberian stalagmite record from Glas resolves the timing of North Atlantic surface freshening anomalies from melting NH ice sheets during the last deglaciation (24 -12 ka BP). After the LGM, a gradual freshening is followed by a series of transient freshening events with onsets at 18.04 (E1), 16.22 (E2) and 15.44 ka BP (E3). Many of these freshening pulses
are present in river-routed time series of the GLAC-1D ice sheet model and meltwater-proxies from sediment cores, but the new Glas record provides a coherent, deglacial history with accurate U-Th dating. Additionally our analysis suggests that for event E1, stalagmite and proxy time series are most consistent with more southerly meltwater routing for the earliest event than derived from river-routed GLAC-1D. Critically, the new speleothem record shows no North Atlantic freshwater impact coincident with the classically cited onset of MWP 1a at 14.6 ka BP, suggesting either that the most rapid pulse to the North
Atlantic event happened nearly 1000 years earlier, or that the freshwater was rapidly advected out of the surface North Atlantic by strong AMOC.

Because this record also resolves regional temperature changes resulting from changes in AMOC strength, the record eluci­dates the evolution of AMOC sensitivity to local meltwater perturbations over the critical period of the last deglaciation. While some transient freshening events (e.g. 16.22 ka BP) trigger immediate albeit brief intensification of the cooling suggestive of
further rapid AMOC weakening, there is a 850 years lag in the response of AMOC to the initial freshening at 18.04 ka. The freshening event at 15.44 ka BP does not trigger further AMOC weakening. These varied responses of AMOC to meltwater in our new record reveal that there is not a ubiquitous response of AMOC to freshening in the North Atlantic. Yet, the similar timing of early meltwater pulses and GLAC-1D suggests that model difficulties simulating AMOC weakenings with GLAC-1D freshwater forcing may reflect challenges in simulating evolving AMOC sensitivity to freshening rather than erroneous timing
of the meltwater forcing. Taken together, these results suggest non-linear dynamics within the coupled ice-ocean-climate sys­tem which invite for advanced transient AMOC sensitivity studies and need to be further evaluated with precise model-data comparisons.

*Data availability.*  Details of the new U-Th data presented in this study is reported in SI Appendix, Table S1. All meta, geochemical and age data for speleothem Glas is further available in SISALv3 submission format (Kaushal et al., 2024) in the ETH Research collection
(https://doi.org/10.3929/ethz-b-000726747), and will be added to the SISAL database with their next release. The underlying model simula­tion data is available from the NERC Centre for Environmental Data Analysis repository (Rome et al., 2022).





## Appendix A: Supplementary Materials

### A1  The Glas sample and its connection to the NISA speleothem archive

Stalagmite Glas originates from the cave 'La Vallina' in Northwest Iberia, which is one of the key caves in the Northwest
Iberian Speleothem Archive (NISA) (Stoll et al., 2022). Glas is a very compact and translucent sample (Fig. A1). Glas has
been selected for this detailed study because of the absence of any signs for non-contiuuous growth and for featuring a
homogenous trace element structure, also when diverging from the main growth axis. Other NISA speleothems were revisited
in the scope of this study, namely Candela and Laura (Fig. A2). Laura confirms the presence of a double-peak freshening
structure before Bølling-Allerød warming occurs (data from Stoll et al. (2022)). A set of new dates on Candela using the
procedure outlined in this study, confirms the presence of a shift around 18.04 ka BP, but has also indicated condensed/ceased
growth from 17.8 ka to 15.6 ka BP (Tab. A1).

### A2  Simulated Temperaure and Precipitation response to a change in AMOC state

A number of studies have previously simulated the effect of changes in AMOC on global climate (e.g., Jackson et al. (2015);
Van Westen et al. (2024)) and the most dominant features such as cooling of the Northern hemisphere and large-scale precipi-
tation changes, such as a weakening of Asian summer monsoon, are robust across models and form the theoretical background
to interpret global archives in the context of AMOC strength. In this study, a quasi-idealised simulation is re-employed from
Romé et al. (2022) featuring a glacial climate state with an oscillating AMOC strength (period ca. 1500 years, see methods
main manuscript), which is triggered by a constant meltwater (MW) flux of 0.084 Sv corresponding to the GLAC-1D ice sheet
history at 17.8 ka BP. From the two strong states defined in Romé et al. (2022), here we selected the more stable *zonal* mode as
the strong mode. When comparing periods of strong and a weak state, the simulation does show the expected global climatic
patterns for a AMOC slowdown, with a temperature reduction of up to 10 K the North Atlantic realm in winter, a southward
shift of the ITCZ and significant changes in winter rainfall patterns (Fig. A3).

### A3  Meltwater time series from sediment cores

In main text Fig. 3, previously published MW time series are presented as simplified blocks for a better visual overview. For
completeness, the underlying time series are presented here in Fig. A4. Note that the event classification in both, Toucanne
et al. (2015) and Wickert et al. (2023) (dataset: Wickert (2023)), are taken from the original publication. For the Baffin Slope
record (Rashid et al., 2012), melt events were set in this study based on the prominent negative $\delta^{18}O$ shifts (shading in Fig.
A4).

### A4  Extended comparison with ice sheet reconstruction GLAC-1D and the derived discharge time series

The GLAC-1D ice sheet model (Ivanovic et al., 2016) was used to derive river-routed regional MW time series for three
key regions (Fig.A5a), which were introduced previously (Romé et al. (2022), see the supplementary information for the



methodology). The three regions are 1.) the GIN Seas (pink shading in Fig. A5a), 2.) Arctic Ocean (green shading), and 3.) Labrador Sea+Gulf of Mexico (GoM) (purple shading). Comparison between the discharge time series derived from GLAC-1D, sediment cores and from the Glas record show significant differences in timing and location of expected discharge, as
featured in Fig. 3 in the main text. These differences likely expose the still existing uncertainties in reconstructing the MW discharge.

For GLAC-1D, the estimated source ice sheet sector for MW pulses ending up in the three regions used in this study (Fig. A5a), is exposed by plotting ice sheet mass loss maps over short time snippets (Fig. A5b-c). However, although this is informative, the timings have to be taken with caution since GLAC-1D is a model, not an ice sheet reconstruction. For example,
the calculation of river-routed MW crucially depends on the ice sheet height and global distribution expected at LGM, which also remains challenging to constrain from data.

Yet, by considering the difference maps in Fig. A5c, for end of Heinrich-Stadial 2 (HS2), strong Southern LIS melt has driven the peak visible in the Laurentide+GoM region and some of Northern LIS/CIS melt also discharges into the Arctic region. Further, some MW that could be visible in Glas maybe also originates from the Southern edge of BIIS. This aligns with
Southern BIIS discharge also being registered by Toucanne et al. (2015), as outlined in the main text Fig.3. In Early-LGM, MW discharges into GIN seas, related to thinning of the ice sheet border between BIIS and SIS, according to GLAC-1D. This slight freshening, close to the Glas site, is consistent and visible as a small excursion in our record. The gradual increase in Late-LGM is reflected in stalagmite Glas, sediment cores and GLAC-1D. For GLAC-1D, most of the discharge seems to originate from southern BIIS, SIS and also Southern part of Laurentide ice sheet. Over this period, proximal MW into Bay of Biscay
seems plausible but also some of the signal could originate from the Southern Laurentide melt. Conversly, the MW sediment cores do not show evidence of MW over this time into the GoM (Wickert et al., 2023). From 18.5-18 ka, GLAC-1D models a rapid melt of Svalbard-Barents-Kara ice sheet (SBKIS), which is dumped into Arctic, which we believed might have at least partially discharged into GIN Seas, or routed to the channel river. In addition, the collapse of Hudson strait, often seen to be connected to enhanced ice berg discharge (HE1), seems to occur from 17.4 k-16.9 ka, reflected in the peak in the purple region
Labrador+GoM. Nothing changes during this time in Glas, and sediment cores, so it might be plausible that this peak could be shifted, i.e. either collapse was later, or melt discharge delayed. The interval 16.4-16 ka shows here mostly the collapse of SBKIS, and is also the only time with ice sheet mass loss in Greenland. This exposes a certain contrast to the general idea to have HE1 then, supposedly sourced from LIS. GLAC-1D 15.4-15 ka shows a peak in GIN seas, originating from SIS and potentially related to Glas 15.4 ka BP. With the beginning of BA, GLAC-1D shows a large retreat of all NH ice sheets, to
accustom for sea level changes, and associated with MWP1a. Interestingly, neither Glas nor any of the discharge proxy time series in this study show a big MW signal there.

## A5 Extended comparison with other global independently dated archives

An extended compilation of global records can be found in Fig. A6. A noteworthy record from another regional cave from Northern Iberian Peninsula is OST2 from Ostolo cave (Bernal-Wormull et al., 2021). This cave features higher elevation and
has nearly double the rainfall compared to La Vallina, indicative of a strong orographic enhancement. Therefore, beside OST2




also showing a shift towards more negative $\delta^{18}O$ likely associated to a changed isotopic signature of the surface ocean moisture source around 18 and 16.1 ka, also the cold air temperature is described to cause negative shifts in $\delta^{18}O$ of the stalagmite, an effect superimposed on the changes in the $\delta^{18}O_{sw}$ seen in the coastal caves. In contrast, in mediterranean caves, such as Meravelles cave (MAAT, SA), the onset of HS1 is synchronous with a shift to more positive $\delta^{18}O$, which in this setting has been interpreted to indicate a decrease in rainfall and resulting deficit in moisture balance consistent with regional downscaled models of precipitation for the region during weakened AMOC. Additionally, stalagmite MAAT shows also an abrupt decrease in temperature, visible in $\delta^{13}C$ (Pérez-Mejías et al., 2021).

## Appendix B: Methods

### B1 Alignment of the old and new trench

Due to the slow growth rate and the presence of abrupt climate changes over the early last deglaciation the sampling strategy for U/Th dating has been optimised by continuously micro-milling the sample with a 50 μm trench along growth layers, measure stable isotopes on these individual samples before combining the remaining powder of about 10 samples to one date. Trace elements and stable isotopes were also measured on a parallel trench and results from the two trenches were aligned using QAnalySeries (Kotov and Pälike, 2018). The two trenches are plotted on top of each other in A7. In all other figures oxygen isotope results are plotted from the same trench as U/Th dates, and $\delta^{13}C_{init}$ are plotted using $\delta^{13}C$ from the U/Th trench corrected with trace elements from the parallel trench.

### B2 Details about $\delta^{13}C_{init}$ correction

Through the correction of the $\delta^{13}C$ ratio for in-cave process of Prior calcite precipitation (PCP), $\delta^{13}C_{init}$ becomes an improved qualitative indicator for temperature above the cave. We follow the procedure of Stoll et al. (2023): the initial Mg/Ca is assumed to be the minimum Mg/Ca ratio over the entire dataset and is used to compute the remaining calcite fraction (fCa) over time. Specifically, the equation to calculate the remaining calcite fraction is $fCa(t) = \frac{min(Mg/Ca)}{(Mg/Ca(t))}$. Based on previous sensitivity experiments for a glacial Northwest Iberian climate and the La Vallina cave setting (Lechleitner et al., 2021; Stoll et al., 2023), the degassing slope $fCa_{slope}$ is set to -8 ‰, the bedrock scaling factor B to 1, and the attenuation factor AF also to 1 for the entire last deglaciation. The resulting equation to calculate the PCP-corrected $\delta^{13}C_{init}$ is $\delta^{13}C_{init}(t) = d13C(t) - fCa_{slope} * log(fCa(t))$.

### B3 Details about age model and changepoint analysis

The age model was constructed from 25 previously unpublished U-Th ages (SI Tab. A1) using the bayesian chronology model BChron (Haslett and Parnell, 2008). In a first iteration, an outlier probability of 0.1 was assumed for each date, and the actual thickness of the combined sample layers was given as a prior to the model. In a second iteration and based on the initial results, the outlier probabilities on the two dates GLA-Z 14.85 and GLA-X 17.55 were increased to 0.4 and 0.8, respectively.



From the ensemble of plausible age models within the 95% confidence interval, we have finally selected the age model that features the highest correlation between computed growthrate and Sr/Mg ratio as the main age model. This was because in this cave setting, the Sr/Mg ratio has been shown to correlate with changes in growth rate (Sliwinski and Stoll, 2021). For the breakpointsd, the changepoints and their age estimates respecting the full 95 % uncertainty derived from the piecewise linear

model are summarized in Table A2 and A3.

## B4    Details about model simulation

The underlying simulation is the experiment 'tfgbj', previously published and made publicly availble in Rome et al. (2022). The simulation was completed using the HadCM3 atmosphere-ocean general circulation model in the BRIDGE (Bristol Research Initiative for the Dynamic Global Environment group) version (Valdes et al., 2017). The model consists of 19 atmosphere

layers (2.5°x3.75°) (Pope et al., 2000), coupled to an ocean model with 20 layers (1.25° x 1.25°) (Gordon et al., 2001), and includes the Moses 2.1 land model and TRIFFID dynamic vegetation model (Cox, 2001). The passive advective tracers were added and run for 500 years in two selected periods, in which the AMOC was strong or weak, respectively.




| Sample Number | 238U (ppb) | | 232Th (ppt) | | 230Th / 232Th (atomic x10-6) | | d234U* (measured) | | 230Th / 238U (activity) | | 230Th Age (yr) (uncorrected) | | 230Th Age (yr) (corrected) | | d234UInitial** (corrected) | | 230Th Age (yr BP)*** (corrected) | | Depth (corrected mm) |
|---|---|---|---|---|---|---|---|---|---|---|---|---|---|---|---|---|---|---|---|
| GLA-I 2.65 | 782.3 | ±0.9 | 720 | ±18 | 1968 | ±50 | 137.3 | ±1.3 | 0.1098 | ±0.0004 | 11050 | ±44 | 11027 | ±47 | 142 | ±1 | 10956 | ±47 | 3.300 |
| GLA-G 3.95 | 837.0 | ±1.4 | 77 | ±10 | 23137 | ±3088 | 137.4 | ±1.4 | 0.1285 | ±0.0004 | 13038 | ±51 | 13036 | ±52 | 143 | ±1 | 12965 | ±52 | 4.525 |
| GLA-F-5.1 | 598.7 | ±0.8 | 1696 | ±37 | 807 | ±18 | 138.3 | ±1.3 | 0.1387 | ±0.0004 | 14131 | ±46 | 14059 | ±68 | 144 | ±1 | 13988 | ±68 | 6.025 |
| GLA-E 6.95 | 796.5 | ±0.9 | 126 | ±14 | 14401 | ±1661 | 139.3 | ±1.3 | 0.1378 | ±0.0005 | 14020 | ±53 | 14016 | ±53 | 145 | ±1 | 13945 | ±53 | 7.700 |
| GLA-D-8.45 | 540.9 | ±0.6 | 68 | ±14 | 18337 | ±3652 | 140.5 | ±1.3 | 0.1399 | ±0.0006 | 14230 | ±64 | 14227 | ±64 | 146 | ±1 | 14156 | ±64 | 9.050 |
| GLA-C 9.65 | 543.4 | ±0.5 | 115 | ±12 | 11043 | ±1140 | 142.3 | ±1.3 | 0.1422 | ±0.0005 | 14460 | ±57 | 14454 | ±57 | 148 | ±1 | 14383 | ±57 | 10.350 |
| GLA-L 11.05 | 375.4 | ±0.5 | 154 | ±14 | 5899 | ±549 | 146.3 | ±1.8 | 0.1467 | ±0.0009 | 14894 | ±97 | 14884 | ±97 | 153 | ±2 | 14813 | ±97 | 11.600 |
| GLA-B 12.15 | 526.6 | ±0.7 | 78 | ±19 | 16913 | ±4146 | 148.9 | ±1.7 | 0.1522 | ±0.0008 | 15447 | ±85 | 15444 | ±85 | 156 | ±2 | 15373 | ±85 | 12.750 |
| GLA-A-13.35 | 386.5 | ±0.4 | 144 | ±11 | 6905 | ±520 | 151.5 | ±1.3 | 0.1565 | ±0.0006 | 15880 | ±69 | 15871 | ±70 | 158 | ±1 | 15800 | ±70 | 13.950 |
| GLA-Z 14.85 | 490.4 | ±0.4 | 118 | ±13 | 11210 | ±1239 | 154.0 | ±1.3 | 0.1639 | ±0.0006 | 16651 | ±70 | 16645 | ±70 | 161 | ±1 | 16574 | ±70 | 15.600 |
| GLA-Y 16.35 | 543.9 | ±0.6 | 328 | ±15 | 4517 | ±201 | 154.2 | ±1.4 | 0.1651 | ±0.0006 | 16778 | ±71 | 16763 | ±72 | 162 | ±1 | 16692 | ±72 | 16.950 |
| GLA-X 17.55 | 667.7 | ±0.7 | 88 | ±13 | 21551 | ±3173 | 155.1 | ±1.3 | 0.1721 | ±0.0005 | 17532 | ±62 | 17529 | ±62 | 163 | ±1 | 17458 | ±62 | 18.225 |
| GLA-W 18.9 | 735.8 | ±0.8 | 62 | ±11 | 33805 | ±5878 | 155.0 | ±1.3 | 0.1722 | ±0.0005 | 17547 | ±62 | 17545 | ±62 | 163 | ±1 | 17474 | ±62 | 19.600 |
| GLA-V 20.3 | 860.4 | ±1.0 | 89 | ±17 | 28385 | ±5490 | 154.8 | ±1.3 | 0.1776 | ±0.0006 | 18141 | ±67 | 18138 | ±67 | 163 | ±1 | 18067 | ±67 | 20.900 |
| GLA-U 21.5 | 490.0 | ±0.5 | 27 | ±12 | 53786 | ±23214 | 154.0 | ±1.3 | 0.1793 | ±0.0007 | 18351 | ±80 | 18349 | ±80 | 162 | ±1 | 18278 | ±80 | 22.125 |
| GLA-T-22.75 | 613.0 | ±0.8 | 74 | ±13 | 25043 | ±4499 | 151.4 | ±1.6 | 0.1837 | ±0.0008 | 18886 | ±91 | 18883 | ±91 | 160 | ±2 | 18812 | ±91 | 23.450 |
| GLA-S-24.15 | 514.0 | ±0.6 | 112 | ±10 | 14153 | ±1239 | 151.5 | ±1.3 | 0.1870 | ±0.0007 | 19253 | ±82 | 19248 | ±82 | 160 | ±1 | 19177 | ±82 | 24.875 |
| GLA-R 25.6 | 445.1 | ±0.5 | 46 | ±10 | 30053 | ±6324 | 153.2 | ±1.5 | 0.1891 | ±0.0007 | 19455 | ±87 | 19452 | ±87 | 162 | ±2 | 19381 | ±87 | 26.225 |
| GLA-P 28.4 | 451.1 | ±0.4 | 21 | ±7 | 70030 | ±23585 | 149.6 | ±1.3 | 0.1963 | ±0.0005 | 20331 | ±57 | 20330 | ±57 | 158 | ±1 | 20256 | ±57 | 29.000 |
| GLA-O 29.6 | 454.5 | ±0.5 | 12 | ±5 | 125285 | ±47103 | 151.4 | ±1.3 | 0.2003 | ±0.0006 | 20755 | ±74 | 20754 | ±74 | 161 | ±1 | 20680 | ±74 | 30.500 |
| GLA-N 32.2 | 408.2 | ±0.6 | 22 | ±4 | 62430 | ±11976 | 150.8 | ±1.5 | 0.2085 | ±0.0006 | 21701 | ±77 | 21700 | ±77 | 160 | ±2 | 21626 | ±77 | 33.100 |
| GLA-M 34.0 | 470.1 | ±0.6 | 24 | ±5 | 68338 | ±13962 | 149.3 | ±1.3 | 0.2117 | ±0.0006 | 22106 | ±77 | 22104 | ±77 | 159 | ±1 | 22030 | ±77 | 34.700 |
| GLA-AA 35.44 | 426.8 | ±0.5 | 64 | ±5 | 23970 | ±2014 | 148.6 | ±1.2 | 0.2172 | ±0.0007 | 22755 | ±81 | 22752 | ±81 | 158 | ±1 | 22678 | ±81 | 36.000 |
| GLA-AB 36.6 | 410.3 | ±0.5 | 24 | ±6 | 62653 | ±16661 | 147.4 | ±1.3 | 0.2212 | ±0.0005 | 23243 | ±69 | 23241 | ±69 | 157 | ±1 | 23167 | ±69 | 37.200 |
| GLA-AC 37.8 | 451.7 | ±0.5 | 60 | ±5 | 27912 | ±2419 | 148.9 | ±1.3 | 0.2245 | ±0.0007 | 23601 | ±82 | 23598 | ±82 | 159 | ±1 | 23524 | ±82 | 38.300 |
| CAN-C +2.06 | 197.3 | ±0.5 | 456 | ±15 | 1343 | ±45 | 230.9 | ±2.8 | 0.1884 | ±0.0013 | 18024 | ±144 | 17969 | ±149 | 243 | ±3 | 17895 | ±149 | 8.3 |
| CAN-G 10.18 | 215.6 | ±0.2 | 431 | ±10 | 1563 | ±38 | 225.9 | ±1.3 | 0.1894 | ±0.0006 | 18210 | ±63 | 18163 | ±71 | 238 | ±1 | 18089 | ±71 | 3.9 |
| CAN-M +18.58 | 187.8 | ±0.4 | 221 | ±6 | 2678 | ±75 | 216.9 | ±2.2 | 0.1908 | ±0.0008 | 18499 | ±95 | 18471 | ±97 | 228 | ±2 | 18397 | ±97 | 5.3 |





**Table A1.** (Table previous page.) **New Uranium and thorium isotopic compositions and 230Th ages for Glas (GLA) and Candela (CAN) samples analyzed by ICP-MS.** U decay constants: l238 = 1.55125x10-10 (Jaffey et al., 1971) and l234 = 2.82206x10-6 (Cheng et al., 2013). Th decay constant: l230 = 9.1705x10-6 (Cheng et al., 2013). *d234U = ([234U/238U]activity – 1)x1000. ** d234Uinitial was calculated based on 230Th age (T), i.e., d234Uinitial = d234Umeasured x el234xT. Corrected 230Th ages assume the initial 230Th/232Th atomic ratio of 4.4 ±2.2 x10-6. Those are the values for a material at secular equilibrium, with the bulk earth 232Th/238U value of 3.8. The errors are arbitrarily assumed to be 50%. ***B.P. stands for "Before Present" where the "Present" is defined as the year 1950 A.D. The depth noted in the last column is the mid-depth of all isotope sample trenches combined for the U-Th dating.





| Age (yr BP) | Age $2\sigma$ negative | Age $2\sigma$ positive | $\delta^{18}O$ |
|---|---|---|---|
| 11382 | 275 | 799 | -3.0156 |
| 14422 | 68 | 189 | -3.3065 |
| 14600 | 131 | 162 | -3.1283 |
| 14788 | 83 | 145 | -3.6251 |
| 15185 | 254 | 110 | -3.9327 |
| 15436 | 85 | 193 | -3.5451 |
| 15993 | 136 | 335 | -4.0323 |
| 16222 | 273 | 224 | -3.2207 |
| 17654 | 94 | 288 | -3.2054 |
| 18036 | 29 | 164 | -2.6244 |
| 20559 | 203 | 169 | -2.1724 |
| 20740 | 82 | 474 | -2.0584 |
| 21701 | 81 | 248 | -2.3183 |
| 22293 | 330 | 400 | -2.1495 |
| 23038 | 206 | 102 | -2.0073 |

**Table A2.** Estimated Breakpoints for $\delta^{18}$O. Ages are given in years before present (yr BP). The uncertainty of the break point estimate integrates the uncertainty from estimating the breakpoint and from the full BChron age model ensemble.





| Age (yr BP) | Age $2\sigma$ negative | Age $2\sigma$ positive | $\delta^{13}C_{init}$ |
|---|---|---|---|
| 12185 | 725 | 463 | -7.5998 |
| 12604 | 835 | 222 | -8.9948 |
| 13061 | 106 | 302 | -8.8653 |
| 13387 | 310 | 288 | -7.9635 |
| 14469 | 62 | 195 | -9.1089 |
| 14767 | 104 | 125 | -5.5894 |
| 16143 | 245 | 284 | -4.4712 |
| 16623 | 194 | 86 | -4.6218 |
| 16876 | 116 | 311 | -3.7553 |
| 17177 | 120 | 343 | -4.4369 |
| 18967 | 144 | 144 | -5.0006 |
| 20609 | 141 | 138 | -4.4498 |
| 21788 | 108 | 193 | -5.9801 |
| 22592 | 282 | 89 | -5.9012 |
| 23038 | 198 | 94 | -6.8444 |

**Table A3.** Estimated Breakpoints for $\delta^{13}C_{init}$. Ages are given in years before present (yr BP). The $2\sigma$ uncertainty of the break point estimate integrates the uncertainty from estimating the breakpoint and from the full BChron age model ensemble.





**Figure A1.** Scan of Stalagmite Glas before cutting a slab for the new trench.





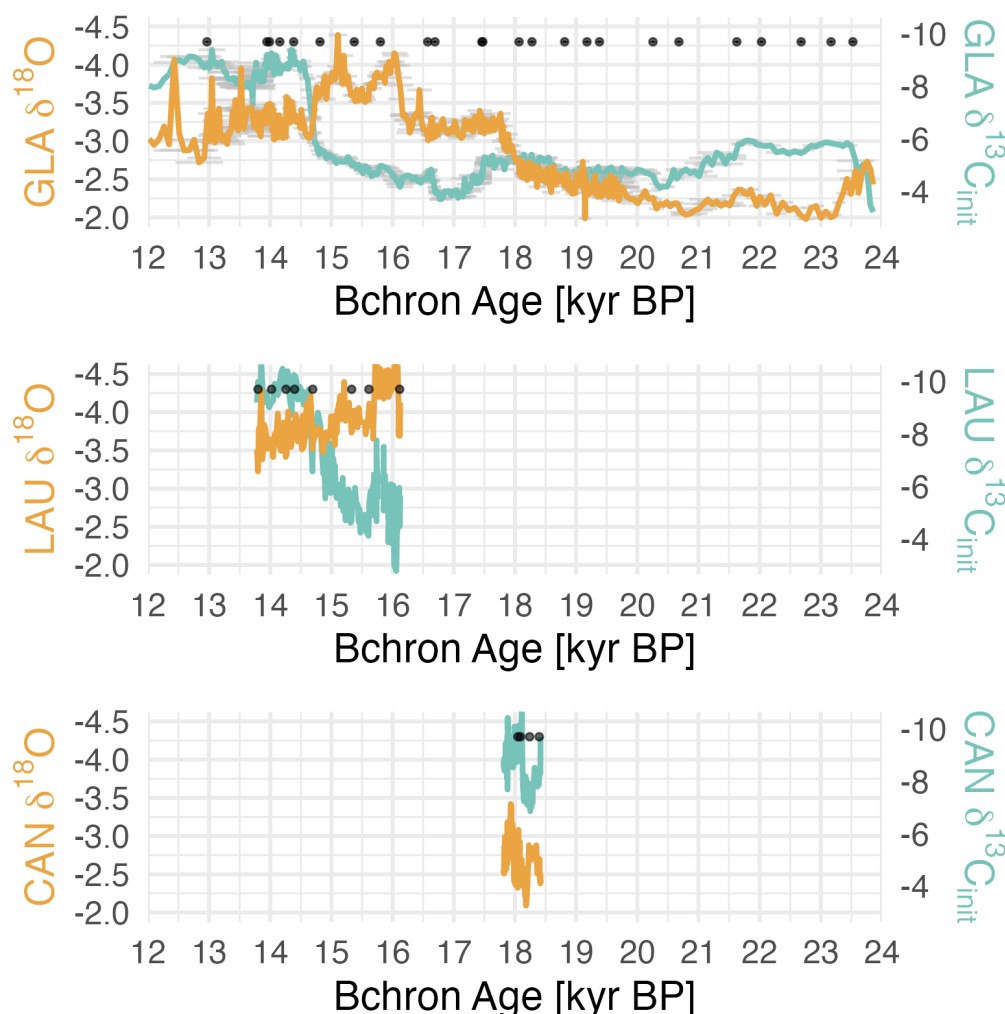

**Figure A2.** Comparison with NISA stalagmites covering segments of the same period: Laura (LAU) and Candela (CAN). Ages of Laura have been previously published in Stoll et al. (2022), ages of Candela are ages on a new trench and are published here in Tab. A1.





**Figure A3.** Global maps of expected environmental changes due to AMOC weakening. Results from the simulation described in Romé et al. (2022), snapshot 17.8 ka. The toprow shows differences (strong-weak AMOC) in surface air temperature (SAT) on the left for winter (DJF) and on the right for summer (JJA). The middle row shows the same analysis for differences in Precipitation (P), and the bottom for total evaporation.





**Figure A4.** Previously published meltwater discharge time series in sediment cores: Baffin slope $\delta^{18}O$ (Rashid et al., 2012)(green), Gulf of Mexico (Wickert et al., 2023)(purple), and from the Bay of Biscay (Toucanne et al., 2015)(orange,blue). Shaded areas are the events featured in the main manuscript Fig. 2. Classification of the events is taken from the original publication for the Gulf of Mexico and the Bay of Biscay. For the Baffin slope record we have defined the events based on the large shifts in $\delta^{18}O$ in the here plotted time series.





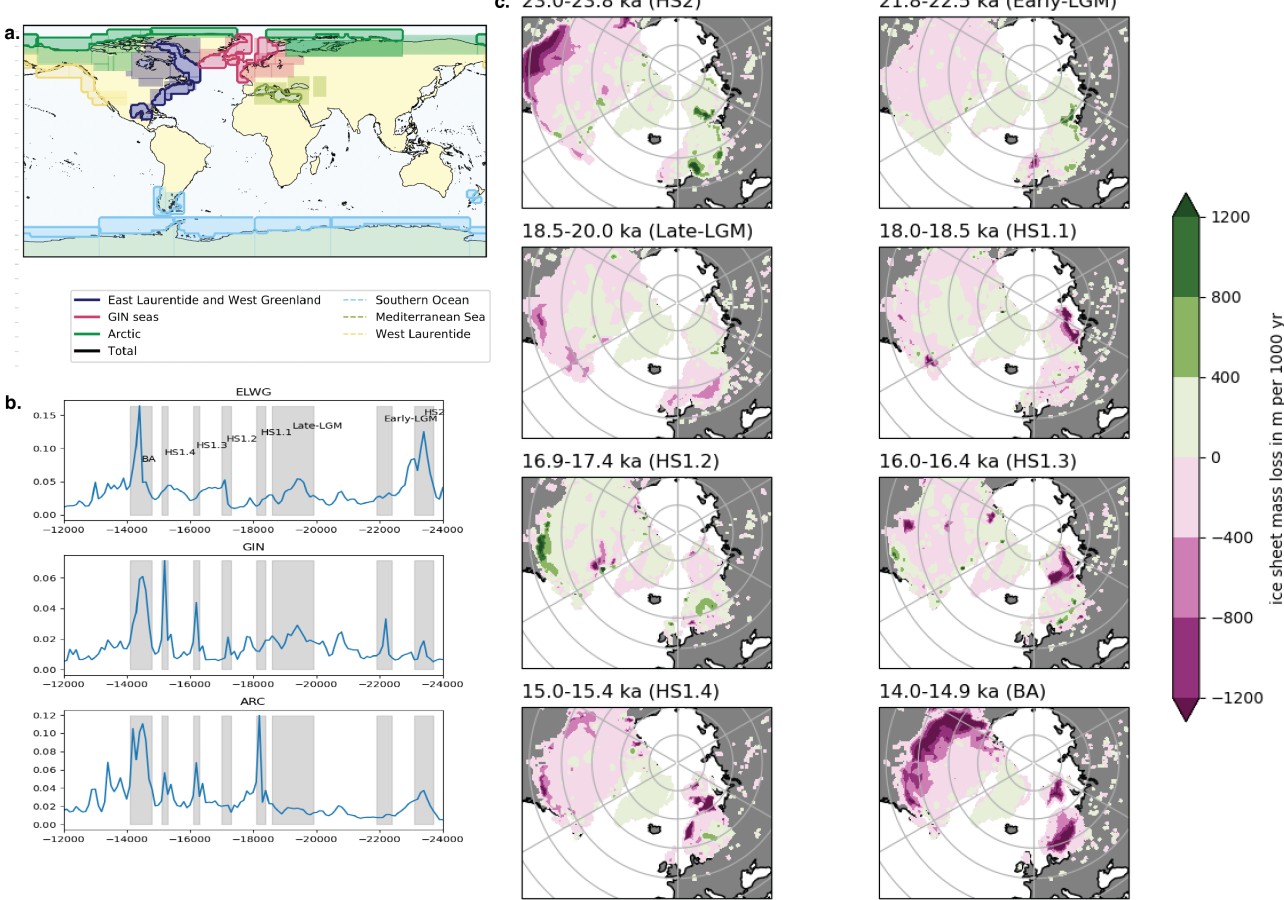

**Figure A5.** GLAC-1D Ice sheet mass loss over selected time snapshots. a) shows a global map including the three regions used in this study East Laurentide and West Greenland (ELWG), Arctic (ARC) and GIN seas, replotted from Romé et al. (2022). In this study, we renamed the ELWG after the drainage ocean bassins and called it Labrador+GoM. b) Detail line plot of the three river-routed time series ELWG, ARC and GIN seas. The grey shadings indicate the time snippets defining name, and start/stop year for the maps in c). c) ice sheet mass loss maps over selected periods with significant peaks in the derived meltwater time series. The ice sheet mass loss has been converted into m per 1000 yr for easier comparison between the panels.



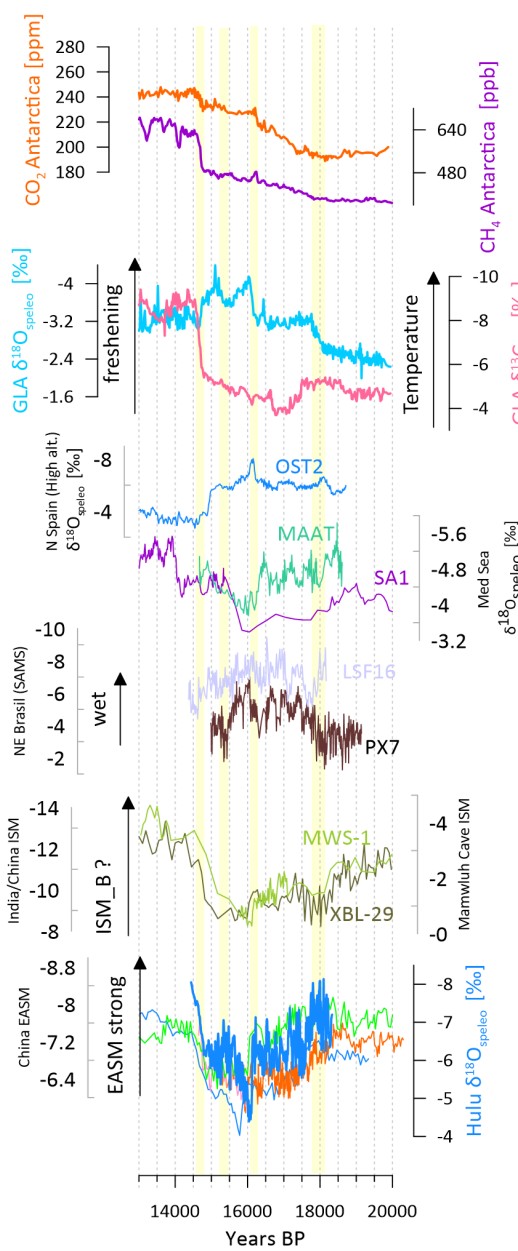

**Figure A6.** Extended comparison with global records. All speleothem data is extracted from SISALv3 (Kaushal et al., 2024) and the Antarctic ice core data from Marcott et al. (2014).



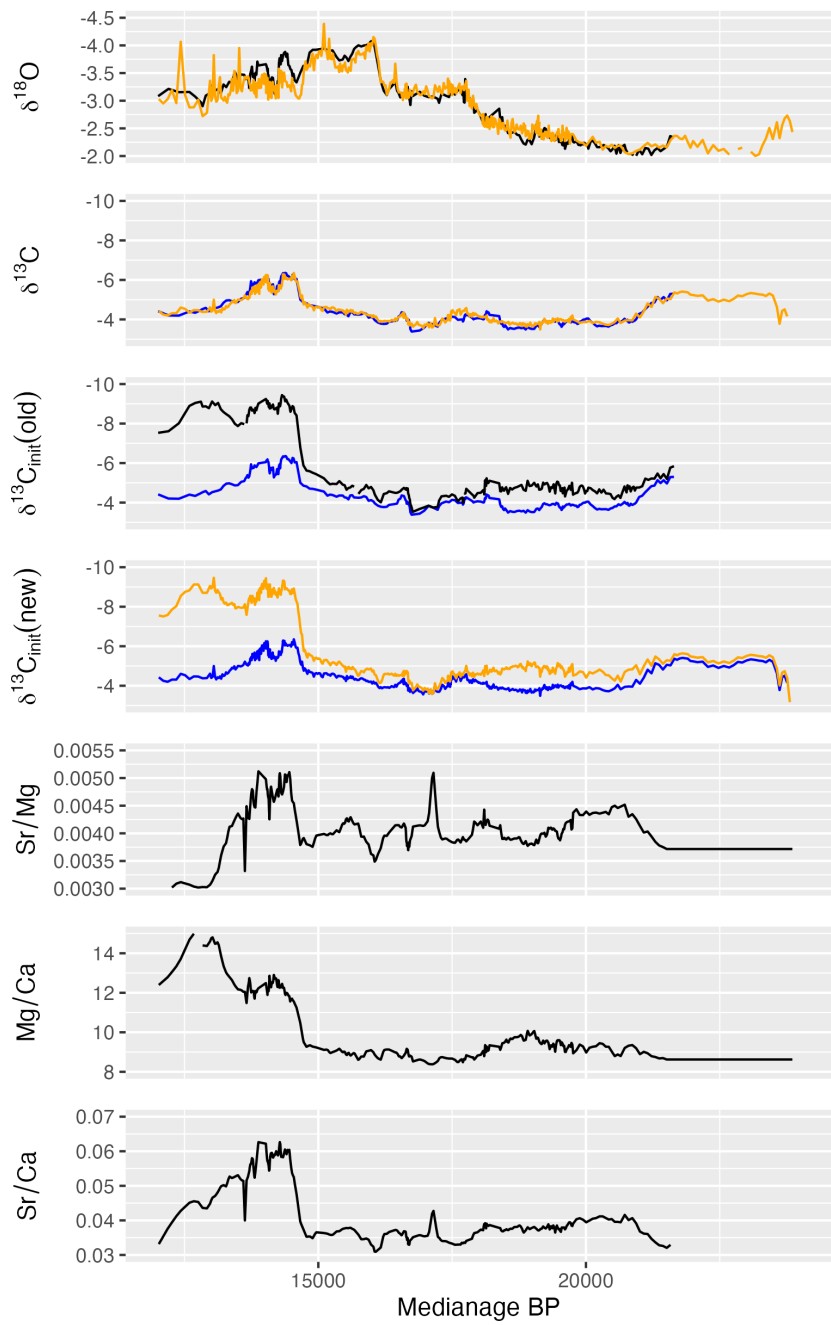

**Figure A7.** Comparison between a prior low resolution trench (black), that was aligned to the new data (orange) with the help of Qanalyseries (Kotov and Pälike, 2018). Trace element data of the old trench was used to correct $\delta^{13}C$ in the new trench.



*Author contributions.* The study was designed by H.M.S.,R.I. and L.E. L.E. wrote the paper with substantial scientific contributions from R.I, L.G., A.L.C.H., and H.M.S. Uran-Thorium dating was conducted by C.P-M and H.C. Fieldwork campaigns were organized by H.M.S.

Sample preparation and geochemical labwork was carried out by L.E. and ETH Zürich technical staff. GCM dye tracer modeling was conducted by L.E., R.I and University of Leeds technical staff. All authors contributed to reading and editing the text.

*Competing interests.* There are no competing interests to declare.

*Acknowledgements.* This work was funded by ETH Zürich base funding and ETH doc.mobility. Additionally funding was provided from the National Natural Science foundation of China (Grant-nr. 42472244). We thank the scientific assistants Janine Schmitter, Sandra Bernegger,

Romain Alosius and especially Lab Manager Madalina Jäggi for their support. Further, we thank Axel Timmermann, James Rae and Andrea Burke for fruitful scientific discussions. OpenAI ChatGPT was used for minor text editing.



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
