# Peer review of "Interplay of North Atlantic Freshening and Deep Convection During the Last Deglaciation Constrained by Iberian Speleothems"

_EGUsphere, 2025_

## Author Comment (AC1)

**Reviewer comment 2**

This is a fascinating and valuable new record, providing important new insights into rapid climate change in its epicentre, the deglacial North Atlantic. The absolute chronology and novel freshwater reconstructions are particularly valuable: they represent a major sampling and analytical achievement and a substantial advance given the impasse in what is possible based on the less precise dates of sediment cores in this interval. The data are well described, nicely interpreted with help from coupled modelling, and well presented throughout.

I provide line by line comments below, the most important of which are starred. The couple of these which I wish the authors to give particular attention to include:

- the interpretation of the stal d13C temperature reconstruction and whether there might be seasonal influences which could lead to some saturation of this proxy, especially given the somewhat different structure it has compared to some nearby Iberian margin SST records;

- the extent to which the stal d13C temperatures necessarily track AMOC – AMOC is one potential control, but there are others;

- the use of onsets vs midpoints, and the naming of events.

On the latter point, I would encourage the authors to use both onsets and midpoints to describe meltwater input events. It's really neat that this amazing archive and your change point analysis can give onsets, but midpoints are probably more commonly used in paleo (as most archives struggle more with onsets), so having both would be helpful and avoid confusion. E.g. the 18.04 ka onset event is the same event as the ~17.8 ka midpoint event seen in many N Atlantic archives; ditto 16.22 vs 16.1 ka.

I note that this is done at 249 and in some other places too, I just think it would be really helpful to do throughout, and would perhaps also be worth the authors considering in their subtitles, abstract, etc., such that where a single number is used, it's one that will be quickly recognisable by the rest of the community – getting folks excited to investigate these more precise determinations of these events in greater detail.

I'm also nervous (as discussed towards the end of the review) about the utility of introducing yet another subset of event names (HS1a,b,c,d ± E1,2,3,4f,4t etc.) into this already crowded space…

Overall I want to stress that I think this is a fantastic contribution and am very supportive of its publication, provided the comments below are dealt with. Indeed please note that the majority of these comments are stylistic or catch minor slips in writing, as I'm excited about this work, so wanted to help polish up its presentation.

Thank you very much for this very supportive review and the great suggestions on how to improve it. We do agree that the current naming of events in our manuscript makes it harder to follow. We will adapt the nomenclature you are suggesting, and will consistently rename the events according to their Midpoint.

Regarding the interpretation of d13C, using your starred comments as a guideline, we will also extend our discussion of the interpretation of stalagmite d13C, including highlighting other possible reasons for Temperature changes besides AMOC decline.

21: writing in general is really nice – so do consider avoiding this clunky phrasing! "cooling in the north, warming in the south, and ..." would be much smoother!

Thanks, we will adjust the sentence accordingly!

45: "the influx of NH MW to the ocean and temperature change..." would be clearer

Thank you, we will adjust the sentence accordingly!

54: measure --> measuring

Thank you for spotting this, we will correct this accordingly.

55: comma after samples,

Thank you, we will add the comma.

57: mid --> midpoint

Yes, agreed – that is clearer phrasing.

58: what is Fiji (other than a Pacific island!) - software?

Although we would be ready to go to the Pacific Island for a nice holiday, we were indeed referring here to software.

We will edit the sentence in the manuscript to include a description and a reference for the software:

The trench drill track was pre-programmed to follow growth layers within the Image Processing Software Fiji (Schindelin et al.,2012), based on prior analysis of microscopic and confocal laser scanning microscopy.

66: say why – explained in caption but good to briefly mention here

Following Main Reviewer Comment 1 by Reviewer 1 we will extend the methodology section to include more details about how and why we used Sr/Mg ratio to select the best fitting age model.

70: need a bit more on analytical approach, for trace elements especially (calibration curve? Sample-standard bracketing? Standards used? Reproducibility of key elements?)

We would propose to extend this section to clarify:

Carbonate powder for trace element anaysis were dissolved in 2% HNO3 and analyzed for trace element to calcium ratios employing an Agilent 8800 QQQ ICP-MS at ETH. Samples and standards were run at Ca concentrations of 400 ppm. Calibration used matrix-matched standards prepared from single element standard solutions to cover the range of trace element to Ca ratios found in stalagmite samples. Our calibration standard composition accounts for trace element impurities in the Ca standard, which were determined using standard additions. Calibration was conducted offline using the intensity ratio method described by de Villiers et al (2002). Analytical drift was corrected with standards run after every 10 unknowns. Precision on Mg/Ca and Sr/Ca ratios is <2% (1s).

de Villiers, S., Greaves, M., Elderfield, H., 2002. An intensity ratio calibration method for the accurate determination of Mg/Ca and Sr/Ca of marine carbonates by ICP-AES. Geochemistry Geophysics Geosystems 3.

81: capitalise Taylor

Thank you, we will adjust this.

86: final clause is jarring - try "with more detailed model description in the Appendix."

We will add the suggested final clause to to the following sentence:
"The original simulation includes 10,000 model years and captures a glacial climate state with an AMOC that oscillates between a strong (relatively 'warm' climate)

and a weak (relatively 'cold' climate) state, with about 1,500 year periodicity, triggered by a constant meltwater flux corresponding to a reconstructed ice sheet history at 17.8 ka BP (GLAC-1D, Ivanovic et al., 2016), with more detailed model description in the Appendix."

91: explain the significance of this region – your key moisture source, right?

The region confined by a red border here is not the moisture source, but the region we used to compute the expected changes in mean climate variables above the climate due to an AMOC decline. To clarify, we will extend the sentence accordingly:

"To study the effect of an AMOC decline on the regional climate above the cave, we have computed the seasonal averages of surface air temperature and precipitation - evaporation, plotted in Fig. 1c-d, for the region confined by an red border in Fig. 1b. "

99: based on fact this is plotted up, it seems more quantitative than qualitative?

The index is not tied to a specific meltwater scenario but rather provides guidance that both the meltwater origin and the AMOC strength are additional controls on how strong a freshening signal is likely present at our site. Thus, we do think of this a s a more qualitative than quantitative index in its present form.
To clarify we will edit the paragraph as follows:

"The results are presented as a qualitative NISA Melt Source Contribution Index (Fig. 1e). The index is not tied to a specific meltwater scenario; instead, it provides a relative estimate of how efficiently meltwater from a given northern ice sheet sector can reach our study site under varying AMOC strengths.
We calculate the index by scaling the model's surface-ocean dye anomalies after 200–300 years of continuous injection (Fig. 1b) using estimated moisture uptake at the site and provide a percentage estimate of three key delivery regions to the full signal at NISA expected for a strong AMOC mode. The uptake estimates are derived from a HYSPLIT analysis based on rainfall at the NISA location El Pindal during 2015–2016.

Figure 1:

in 1e axis label change GIN Seas to GIN Seas and NE Atl.  I also don't understand the use of % with strong AMOC 100%, as doesn't seem like the red bars would add to give 100%?  Is there another melt source or what am I missing?

We will change the Region name to GIN Seas and NE Atl., and will update the Figure to scale to 100 %.

in caption: bassins ◊ basins; Northern hemisphere ice sheet's ◊ Northern hemisphere ice sheet; "petrol" is not a widely known colour – "grey/green" or "light green" would be better

We will change the color name to "light green".

120: sensitivity index or contribution index as in 1e caption?  Or are these deliberately different?

Thank you for spotting this inconsistency. It should be Melt Source Contribution index.

124: would delivered to the NE Atlantic and GIN Seas be more accurate here?  I realise that what's delivered to NE Atlantic is then effectively transported to GIN Seas, but melt itself may have come in to NE Atlantic to the south of GIN Seas.

Thank you for the suggestion. Yes, we can write this more clearly by stating: "Furthermore, we can identify that under the weak AMOC state, the tracked NISA source region more strongly accumulates meltwater of Eurasian origin delivered to the GIN seas and the Eastern North Atlantic, whereas meltwater discharged by the Laurentide ice sheet to the western North Atlantic more strongly accumulates in the tracked region under strong AMOC conditions (Fig. 1b)."

Figure 2:

Don't see yellow dots!  And what is significance of open vs closed diamond symbols?

Thank you for highlighting this mistake. In the present figure, the U-Th dates are marked as diamonds. We will update the figure caption to include referring to the correct shape and add a sentence about the open vs closed diamond symbols

"Open and filled symbols were chosen to simplify retrieving the width of a specific dating hole by comparing directly with the purple vertical errorbar indicators."

168: the presence of meltwater is confirmed, but Scandinavian source is then an inference based on the combination of this melt evidence with the eNd data, right?  So slightly rephrase.

We will edit the sentence to:
"This progressive freshening confirms the presence of MW in the eastern North Atlantic, for which a source could be meltwater from the Scandinavian ice-sheet (SIS) as previously suggested by Nd isotope provenance of detrital minerals delivered into the Bay of Biscay.

173: I don't obviously see these other regional records in the SI – please specify figure.  The compilation in A6 is useful, but these are more globally distributed rather than regional.

We will refer more clearly to the record OST2, which is available in the supplementary.

179: as written, it's a bit unclear how the SBKIS links to the statement above about SIS and BIIS separation.  Are these the same event e.g. the SIS and BIIS separate due to the collapse of SBKIS which previously joined them?  Slight rephrasing to more clearly express how these different pieces fit together would be helpful.

We will here more generally state that event-like retreat of EIS at this time with meltwater discharging into GIN Seas and NE Atlantic are likely causing the freshening signal at this time in our record.

184: again GIN Seas or GIN Seas and NE Atlantic?

We will edit to GIN Seas and NE Atlantic.

184: sustained low d18O values

 We will edit the sentence.

190: onset of the "15.44" event really looks a bit younger on the figure – more like 15.3. Please check!

We will provide updated nomenclature using the midpoints of transitions. Further, we will add the breakpoints to this figure 3 on a separate axis for better comparison and identification.

Related, given the centennial scale of your data, it would be nice to have 100 year tick marks.

We will add the the 100 year tick marks on the x axis.

202: use of "this" is confusing, as ice sheet shrinkage was not previously being explicitly discussed.

We will replace this with "continued".

***203: this is an interesting idea but would be good to flesh out a bit more.  One important point is that – based on the analyses of model output presented above – it seems possible that the same degree of ice sheet variability might be expressed as a spikier signal if there is strong AMOC, as successive melt signals would be quickly dispersed, leading to a spikier record, even with little change in ice sheet behaviour.

Related – and tying in to interesting discussion of MWP1a below, could there be continued MW through BA onset, but d18O signal shows an apparent drop off, due to switch in AMOC regime?  In other words, the change in d18O at 14.7 ka, which might at first glance be interpreted as a decrease in MW input, may actually be a signal of a restart in AMOC which dissipates the MW signal, even if the same rate of MW input persists.

I see some of this is discussed at line 260 – nice job!  However I think still worth bringing out this nuance here and see also some of the further discussion in review comment there too.

We agree that it would clarify the message that  a strong AMOC disperses the freshening signal much better, with consequences on what can be seen in the Glas record. We will improve the connection between the discussion in lines 203 and 260 to clarify the the restrengthened AMOC as a complimentary hypothesis for the spikier signal in this paragraph and provide more nuances to the MWP1A interpretation.

207: as mentioned above, this is another place where use of onset is potentially confusing: 15.44 seems a long way from ~14.5 ka dates for MWP1a; whereas if you were to say "centred on 15 ka" or "extending from 15.4 to 14.7 ka" that would be easier to follow and to potentially reconcile with MWP1a.

See also Coonin et al., 2025 NGS for recent re-evaluation of MWP1a age constraints.

 We agree to clarify the language as suggested . Thank you also for the reference, which we will gladly include.

***217: Here and where d13C temperature proxy is introduced above: could the temperature proxy get saturated towards cold temperatures?  There's surprisingly little variability within HS1 compared to, say, %NP, Mg/Ca, or Uk37 records in the North Atlantic.  Might this be explained by the coldest seasonal temperatures not really being recorded due to freezing? The d13C proxy would thus bottom out, recording low respiration in the unfrozen shoulder seasons, but not capturing peak winter cooling?

We are convinced that the amplitude of the d13C anomalies are attenuated by the sampling technique.

 We propose to provide the following clarification about the proxy seasonality in section 2.

The d13C of soil CO2 follows the Keeling mixing line (eg as shown in cave monitoring study of Kost et al, 2023) meaning that the soil and dripwater d13C is actually more sensitive to soil pCO2 at low soil pCO2 (see also Lechleitner et al. 2021) .  Because this slow growing stalagmite does not feature seasonal laminations, it is difficult to independently assess changing seasonal biases in stalagmite growth. However, the lack of seasonal laminations (and high DCF) suggest a matrix fed system which at the depth of the cave passage (>20 m) may have continued to have drip flow through the cold season because there is a strong attenuation of the seasonal temperature cycle below 5 m and during deglacial AMOC shutdown the temperature at the base of the soil remains above freezing all year (e.g. Tapia et al., 2025 Fig. 8).

We propose to increase the scale of the d13C axis in Figure 2, and to provide the following clarification about signal smoothing in section 2:

The width of the sample trench has the advantage of providing perfect coupling between isotope records and chronology, but due to the slow growth rate leads to smoothing of the d13C signal, which attenuates the magnitude of centennial scale excursions.

- Kost, Oliver, Saúl González-Lemos, Laura Rodríguez-Rodríguez, Jakub Sliwinski, Laura Endres, Negar Haghipour, and Heather Stoll. "Relationship of seasonal variations in drip water δ 13 C DIC, δ 18 O, and trace elements with surface and physical cave conditions of La Vallina cave, NW Spain." *Hydrology and Earth System Sciences* 27, no. 11 (2023): 2227-2255.

Lechleitner, Franziska A., Christopher C. Day, Oliver Kost, Micah Wilhelm, Negar Haghipour, Gideon M. Henderson, and Heather M. Stoll. "Stalagmite carbon isotopes suggest deglacial increase in soil respiration in western Europe driven by temperature change." *Climate of the Past Discussions* 2021 (2021): 1-25.

Tapia, Nicolas, Laura Endres, Madalina Jaggi, and Heather Stoll. "Accelerated phosphorous leaching during abrupt climate transitions in a temperate Atlantic ecosystem in Northwest Spain recorded by stalagmite P/Ca variations." *Biogeosciences* 22, no. 22 (2025): 6861-6875.

Figure 3:

Don't need simplified at both start and end of the sentence!

We will replace the latter with "original data available in Appendix".

220: unclear which "rapid decline" is being referred to: the short blip at 17.0 ka or the gradual cooling from 19-17 ka?

With the rapid decline we were referring to the "short blip". We will clarify this here by using the new midpoint nomenclature.

224: though would be interesting to compare to what structure there is within the d15N data (e.g. Buizert et al., 2014) – muted but does seem like NGRIP may share some similar structure to NISA d13C

We appreciate the suggestion to examine the Tsite reconstructions of Greenland temperature that result from modeling firn processes and d15N. We agree these records provide clear agreement with the onset of the Bolling/Allerod warming which is also clear in the plotted d18O NGRIP curve. However, we refrain from making detailed comparisons with the Tsite reconstructions during the Heinrich where the modeled amplitude of variability in Tsite is very small relative to the uncertainty.

227: agree it represents a genuine lag – but "in response time" implies that cooling would always follow meltwater input, which may or may not be true (as is discussed below). So suggest rephrasing.

Thank you, we agree and are happy to remove "response time".

247: and by Barker et al., 2015 %NP vs IRD records

Yes, thank you for the suggestion.  We will add this reference.

260: see comment above at 203 – I think it would be good here to be slightly more specific. It's a reduction of meltwater abundance at the surface, though might not be a change in the rate of meltwater input from ice, just its accumulation and persistence in the surface.

Yes, we appreciate the suggestion to again incorporate here the important findings from the model simulation and will do so in the revision.

266-268: this is interesting and I think would benefit from a bit of elaboration and nuance, as in detail the structures of these records are quite different:

We appreciate the suggestion and will expand the discussion slightly, and acknowledge that the amplitude of centennial events in the NGRIP d18O differs slightly from that in the Tsite derived from d15N.

Line 268 mentions slowed growth rate, but this only kicks in around 13 ka, so prior to that should be reasonably well resolved?  Perhaps a bigger issue is the gap between age control points?  I wonder if the age uncertainty from the age model (i.e. the shaded band from Figure 2) could be shown as a line next to the age control points in this figure to give a sense of this uncertainty?

The growth rate declines a bit earlier already, which is related to the larger spacing of the age control points. We will make more clear during the revision that we advise to use the Glas record only into BA but not for younger segments.

269-271: this is a really interesting point, but I'm not sure I quite follow it or see the events being referred to.  Could these be spelled out or shown in more detail?  Are you meaning that the events at 14.4 and 14.0 ka appear to have MW then cooling, whereas this is not clear in event at 13.6, though perhaps is again at 13.2?  Also in the 14.4, 14.0, and 13.6 ka events, it seems that there's an initial warming coincident with the initial meltwater input – would you read anything in to this?

We agree that this would be a nice interval to study but we do not believe that the resolution of Glas is sufficient to resolve and discuss these events in detail. We would prefer to keep this statement broad but will clarify the events by providing a bracketed time interval within BA.

276-277: potentially confusing to introduce both HS1a,b,c,d and E1,2,3,4 terminology.  Can do so if you wish, but could consider using the same system throughout.  (And note that using "Events" within the Heinrich stadial, which are distinct from the Heinrich Event(s) themselves, adds an extra layer of potential confusion!  E.g. I don't think the community will adopt thinking about Event 1 within Heinrich Stadial 1, which is distinct from Heinrich Event 1...).  Might be easiest to just use the mid-point ages...

Agh, now see that in Figure 4 they also come with f and t flavours! Think this is probably just too complicated… Can highlight freshening and warming or cooling events, perhaps using different colours for each and a letter at the top, but think then easiest to just use their timings and to refer to them descriptively within the text e.g. "the freshening event centred on 17.8 ka…"

Thank you for highlighting this issue. We will follow your suggestion and edit the manuscript throughout to a consistent midpoint nomenclature.

282: confused here about what is being referred to, as the pink bar from 18-17.6 ka seems to be associated with a minima in d18O so a negative shift… if you mean a different feature (like the subsequent positive shift in Hulu at 17.8-17.5ka) then spell out more clearly in text (i.e. following the onset of the meltwater event beginning at 18.04 ka, a positive shift is seen in Hulu from 17.8-17.5 ka) and consider how you use and place the shaded bars (±some dashed lines) to help highlight features in your record vs Hulu and others.

We will revise Figure 4 to feature records from more regions and improve the visualisation of common features between our record and other global proxy data.

Figure 4 has two panel a's

We will adjust the labeling in our revised Figure 4.

**284: I'm not sure how strongly this temperature change should be referred to as the most abrupt AMOC reduction. Change in AMOC is one possibility, but so is change in sea ice, or non-AMOC circulation changes. It's plausible that AMOC may have a distinct structure to this particular temperature record, especially given that other nearby SST records have slightly different structure within this interval.

We will add some further clarification and references to underscore that in the eastern North Atlantic, around the Iberian peninsula, the cooling from AMOC weakening is a very robust signal seen in all models in intercomparison studies. This is a very solid basis for the identification of AMOC weakening. In the revision, we will expand our discussion to review the role of sea ice as an important feedback to cooling in the North Atlantic region.

The Iberian Margin SST records (the main nearby temperature records we assume the reader refers to) provide useful overview of temperature resolution and context but because of the dynamic sedimentary environment multiple sediment core records (even with the same proxy) exhibit some diversity in the structure (Rodrigues et al 2010).

Rodrigues, T., Grimalt, J. O., Abrantes, F., Naughton, F., & Flores, J.-A. (2010). The last glacial–interglacial transition (LGIT) in the western mid-latitudes of the North Atlantic: Abrupt sea surface temperature change and sea level implications. *Quaternary Science Reviews*, *29*(15–16), 1853–1862. https://doi.org/10.1016/j.quascirev.2010.04.004

286: beginning at 16.22 ka and centred on…

We will update the text using the midpoint nomenclature, noting here centered on 16.1ka.

295: northward?

Thank you for spotting this mistake.

304: again, I'd be a bit careful with this, as although no abrupt cooling at this time is seen in the stal d13C, some of the SST records from e.g. Iberian Margin do show more structure within this general interval.  So the stal d13C is not the only/last word on temperature change in this region and time!

We will add a clarification that, given the importance of this novel finding from the GLAS record, future studies should seek to test the phasing in other high resolution (ie non-bioturbated) records with high precision absolute chronology.

305: interesting idea, given that even if not related to AMOC, many ideas about atmospheric reorganisation would invoke temperature change.  What might you suggest as alternative?  Ice sheet height?

We were speculating here global reorganization of atmosphere patterns related processes such as expansion of southern hemisphere westerlies, shift in hadley cells and/or ice sheet height. We will specify this in the manuscript and provide the according references, such as Trombini et al., 2025.

315: interesting that the 15.4 ka timing seems to also correspond to Antarctic melt water input, based on Li et al. (2023), Nat. Comms.

This is interesting and further highlights the multiple ways meltwater is entering the global ocean over the last deglaciation. However, as we are focusing on the North Atlantic realm in this manuscript, we opt to not include this record into our comparison figure.

319-322: again, encourage authors to reconsider Event terminology.

We agree.

323-324: see comment above – based on the results presented here, especially the model output, it seems possible that AMOC onset could mask a MWP1a signal, at least partially,

by dissipating the d18O freshwater signal.  I realise that you go on to mention this at line 325, but I think it's worth being really careful with this initial statement, as many folks will just read the conclusions and may not grasp this potentially important detail.

We will reformulate these sentences in the conclusion and will be refering explicitely back to our modeling results to strengthen the interpretation that a MWP1a signal could be masked in our record.

330: as per comment at line 304, I'd be careful making an unambiguous link between AMOC and NISA d13C.

We will add a general remark of caution here.

Figure A4. Updated d18O in labels to proper formatting

We will update the labels.

---

## Author Comment (AC2)

**Reviewer comment 1**

The manuscript by Endres et al. presents a high-resolution, Th/U-dated speleothem record from northwestern Iberia spanning 24-12 ka BP. The analyzed proxies are used to reconstruct North Atlantic surface ocean freshening ($\delta^{18}O$ values) and regional temperature changes ($\delta^{13}C$) during the last deglaciation. The authors identify major freshening events during Heinrich Stadial 1, and claim that the initial cooling response lagged the first meltwater pulse by approximately 850 years, suggesting evolving AMOC sensitivity to freshwater forcing. Overall, the study provides interesting new constraints on the temporal relationship between ice sheet meltwater discharge and Atlantic Meridional Overturning Circulation strength. The manuscript is overall well-written and the analytical methods are sound. Also the interpretation of the proxies is based on an intensive work of the research group in that area. However, I have some comments on the description of statistical aspects including propagation of uncertainties as well as the discussion of the regional relevance of the results. Overall, the line of arguments could benefit from restructuring the discussion by re-integrating parts of the extended appendices back into the main text. It is a bit exhausting to repeatedly having to switch back and forth between main text and appendix, also given that the main text is not so long.

Thank you very much for the assessment and the detailed review. We are happy to include your comments and suggestions to the final manuscript. Specifically, we will extend the description of the statistical aspects of the work in the sections outlined below. Following your suggestion, we will also extend / re-arrange / re-write the discussion section 3.5 Impacts beyond the North Atlantic Realm to include the information from the appendices.

**Other comments:**

1. **Age model.** The authors state that they have optimized the age model by using the Sr/Mg ratio in the stalagmite. Please discuss a little but further why this approach is valid at this coastal location, where one could expect a significant and varying influence of sea spray, that could affect the Sr/Mg ratio as well. Also provide more details what has been exactly done and how the age model has been optimized. Have you smoothed the Sr/Mg ratio prior correlation analysis with growth rate? What was the highest correlation coefficient? How does this approach influence the finally given uncertainties of the age model (and then, the uncertainties of the break/changepoints).

Thanks for the prompt to clarify this issue. We will extend the section 2.2 U-Th Dating and Age Model to address these questions, implementing the information given here below.

Related to using the Sr/Mg ratio:

The delivery of sea aerosols is insensitive to the changes in distance from the coast in the range between the current (43.5km from coast) and estimated glacial maximum distance (8

km from coast) of the cave (Kost and Stoll, 2023 ). Median modern dripwater Na concentrations in interior cave sectors where GLAS was collected are 5 ppm (Kost et al., 2023).  For the minimum stalagmite Mg/Ca at 9 mmol/mol (DMg of 0.0225) this Mg from marine aerosols would constitute 3 to 6% of the dripwater Mg for open system dissolution in the range of 600 to 8000 ppmv  $CO_2$, leading to variations in the dripwater Mg/Ca and Sr/Mg of about 3% between glacial and interglacial endmembers, a very small variation compared to the measured Sr/Mg range of 0.002 - 0.006 (a 40% increase) in this sample. Further, Sr/Mg ratio has shown to be covarying with growth rate also in other samples of the NISA archive (Sliwinski et al., 2023).

Related to the age model:

The age-depth model has been computed per the algorithm of Haslett and Parnell (2008) using the package BChron with 50'000 iterations and by explicitly providing the dating sample thickness as an additional constraint (as this is known because of our drilling method) and all samples have been assigned an outlier probability of 0.1. The age uncertainties stated in the paper are the 95% CI output of this statistical model and we refrained from further constraining these, thus this remains a conservative estimate. However, instead of using the statistical median model as our author age-depth model we have selected from the full ensemble of plausible age-depth model created by BChron the one model featuring the highest correlation with the Sr/Mg ratio, following the rationale outlined above. By selecting the alternative age model, abrupt changes in growth rate were mitigated and the pearson correlation coefficient between Sr/Mg and  has improved from 0.27 (BChron Median model) to 0.60 (SrMg optimised Model).

We will provide a figure in the supplementary where the growth rate of the two age models can be compared directly.

Kost, Oliver, and Heather Stoll. "Marine aerosols in coastal areas and their impact on cave drip water– A monitoring study from Northern Spain." *Atmospheric Environment* 302 (2023): 119730.

Kost, Oliver, Saúl González-Lemos, Laura Rodríguez-Rodríguez, Jakub Sliwinski, Laura Endres, Negar Haghipour, and Heather Stoll. "Relationship of seasonal variations in drip water δ 13 C DIC, δ 18 O, and trace elements with surface and physical cave conditions of La Vallina cave, NW Spain." *Hydrology and Earth System Sciences* 27, no. 11 (2023): 2227-2255.

Sliwinski, J. T., Oliver Kost, Laura Endres, Miguel Iglesias, Negar Haghipour, Saúl González-Lemos, and Heather M. Stoll. "Exploring soluble and colloidally transported trace elements in stalagmites: The strontium-yttrium connection." *Geochimica et Cosmochimica Acta* 343 (2023): 64-83.

**Breakpoints/Changepoints**. The outcomes of the study rely strongly on statistical time series analysis and the correct propagation of uncertainties. It is however unclear to me, how the age model uncertainties have been propagated into the uncertainties of the break/change point analysis. The resulting change/breakpoints are also only given in a table. It would be beneficial to visualize the timing of the events in the stalagmite also in Figure 3.

Please also improve the quality of Figure 3, the discussed "drops" or other events in the δ¹³C record are not visible in the Figure, possibly because the y axis is too small. Please improve and make it easier for the reader to follow your arguments...

Thank you for the comments and suggestion to clarify.

We will extend the explanation in the method section and will provide the Analysis R script in a public zenodo repository, to clarify that the age uncertainties have been propagated by conducting the Breakpoint analysis not only on the main age model but on the full BChron ensemble. As outlined above, our age model uncertainties are a conservative estimate and we are fully propagating the uncertainty here. This propagation makes the uncertainty of the changepoints relatively large, and thus, again, is a very conservative estimate. We will edit Figure three to enlarge the y axis for the d13C_init plot.

2. **Meltwater model results.** I am not a modeler and after reading the manuscript the relevance of the meltwater/tracer modeling is still unclear to me. I feel like the main conclusions could also be drawn without all this effort..? If I could follow the manuscript correctly, the model results are mainly part of the methods section, and there is only a limited discussion of model implications in the results/discussion. Could the results be helpful to explain a mechanism why there is a 850y lag between meltwater and temperature? Overall, the integration of the model results into the results/discussion section should be improved and the relevance clarified for non-experts in that field.

Thank you for your comments. and the suggestion to better integrate the way the modeling results improve understanding of the proxy record.

We propose to more clearly highlight throughout the discussion the aspects of the interpretation which are aided by the illustrated model experiments. One key takeaway from the model experiments is the documentation that under periods of stronger AMOC the freshwater d18O signal is quickly removed from the surface ocean, leaving a very weak fingerprint. This supports our interpretation that a significant freshening signal from MWP1a may not be salient in the stalagmite proxy record because of strong AMOC at this time.

Regarding model support for a delayed AMOC response to freshwater, we revise the discussion paragraph L.234-243 to provide a clearer theoretical background why a delay in the AMOC response is consistent with our theoretical understanding of AMOC stability, supported by published model results. One high resolution GCM model found AMOC collapse occurred 1750 model years after the onset of a gradually increasing North Atlantic freshwater forcing (van Westen and Dijkstra, 2023). Simulations in HadCM3 (Rome et al., 2025) under glacial boundary conditions exhibit lags and oscillatory response to a constant freshwater forcing. These are consistent with AMOC theory (e.g. Barker et al 2021). We will clarify that the dye trace models described in this manuscript do not explicitly

investigate the time lag which is already discussed in the underlying simulations by Rome et al 2025.

van Westen, René M., and Henk A. Dijkstra. "Asymmetry of AMOC hysteresis in a state-of-the-art global climate model." *Geophysical Research Letters* 50, no. 22 (2023): e2023GL106088.

Romé, Yvan M., Ruza F. Ivanovic, Lauren J. Gregoire, Didier Swingedouw, Sam Sherriff-Tadano, and Reyk Börner. "Simulated millennial-scale climate variability driven by a convection–advection oscillator." *Climate Dynamics* 63, no. 3 (2025): 150.

Barker, Stephen, and Gregor Knorr. "Millennial scale feedbacks determine the shape and rapidity of glacial termination." *Nature Communications* 12, no. 1 (2021): 2273.

3. **"Regional/global discussion".** The comparison with other records is very limited and could be strengthened. On the regional scope, only 2 other speleothem records from Iberia and the Mediterranean are mentioned and only in the appendix. It is unclear to me why the discussion then only focuses on comparing to records from the EASM region, where the comparison then turns out to be not as conclusive (compare L296ff). I strongly suggest to revise this section, and focus on a more integrative approach comparing first to records from the wider North Atlantic realm in more detail. There are e.g., some high-resolution records from across the Americas (for example Travis Taylor et al., 2025, Strikis et al., 2015, and others) that are much closer to the centers of action in the North Atlantic. There are also probably some more from Western/Central Europe... (e.g., NALPS, Li et al., 2021, Luetscher et al., 2015, ...). Incorporating more records from the wider Atlantic region could support the discussion of potential "atmospheric responses", that is at this stage only briefly mentioned in L252 or 290, but I regard this is as not irrelevant in this discussion. Also, in a subnote, I wonder about the absence of a drop into the Younger Dryas at the end of the record. I know this is not the focus of the manuscript but it still makes me wonder... Many other stalagmite records show a very sharp and clear drop in $\delta^{18}O$ values around 12.9ka BP (e.g., Cheng et al., 2020, Affolter et al 2019, Li et al., 2021 ...) , but this one, that should presumably be very sensitive to meltwater and temperature in the north Atlantic realm, does not really show a clear signal, at least not in $\delta^{18}O$... What does this mean?

We thank the reviewer for prompting us to examine which records are included in compraison in the main text figure and the supplementary figure.

First of all, we will clarify in results, that the focus of interpretation on the new record is from the LGM through the early Bolling Allerod; by the onset of the Younger Dryas, stalagmite GLAS has a significant drop in growth rates, reducing significantly both the resolution of proxy data due to smoothing, and the precision of chronological constraints.

We will clarify in the discussion that we focus our comparison on the records which cover this full period (e.g. 20 ka to 14 ka) with comparable precision in chronology as the GLAS record.  Because it is not the purpose of this paper to review and re evaluate the diverse factors encoded in global stalagmite d18O records nor assess the impact of AMOC on these

processes, we also will clarify that we focus comparison on records from locations where a clear link with AMOC has been previously described through models, theory, or data, including during AMOC weakening : the cooling in western Europe (NALPS Speleothem (Li et al., 2021; Luetscher et al., 2015), the southward ITCZ migration in South America (Brazil Speleothem Paixao Cave (Strikis 2015):, and the East Asian monsoon response (already Hulu Cave shown in Figure 4). We hope that our new record will provide an important North Atlantic reference for future studies to test how atmospheric processes in other regions may be affected by AMOC, but such a discussion is beyond the scope of this paper.

-

**Minor comments along the text:**

L65 Please provide more support that Sr/Mg is indeed only growth rate and not related to sea spray

We will extend section 2.2 to convey this information, as outlined in our reply to main comment 1.

Figure 2 What do the open and filled symbold mean? What is the correlation between Sr/Mg and growth rate?

We will add to the Figure caption for clarification: Open and filled symbols were chosen to simplify retrieving the width of a specific date on the age-depth axis by comparing with the purple estimates. The pearson correlation coefficient between Sr/Mg and growth rate is 0.6.

L171 does the number 382 years correspond to the length of the transition? Please indictae the timing and uncertainties in the Figure, it is hard to see for some of the breakpoints where they are exactly identified.

Thank you. We will add the breaks and midpoints to Figure 3.

L189 Also here, I am not sure which "smaller transient event" is meant. Possibly a second plot that zooms only into the most interesting section of the record would be helpful?

Following the suggestion of the 2$^{nd}$ reviewer we will update the nomenclature and name the events by their age midpoints. The "smaller transient event" will be named FE_16.45k, and as this indicated in Figure 3.

Figure 3 please include time markers of identified events in Figure 3. I also cannot see the temperature changes in d13C as mentioned in the text... The relevance of the maps in the uppermost panel are unclear to me.

Following the previous responses, we will add time markers and event names to Figure 3. We will also update the map to show ice sheet mass loss (like in the appendix) directly. We believe that this will improve the visual message that the key interest is in enhancing the understanding of melting from different ice sheet sectors.

L225 I cannot see a "significant cooling" in d13C at that time in Figure 3, I think the y-axis not large enough to really see this drop? A clear marker would be also helpful.

We will add a marker noting the event. We will also enlarge the y-Axis of d13C_init.

L226 what is the uncertainty of the 850 years?

We will provide an estimate for the uncertainty.

L231 Again, unsure to which specific feature this refers to

We apologize for the confusion and will provide the updated nomenclature with naming by event type and their midpoint. Thus, The cold event referred to here will be called TE_17.01k. This event will be annotated in Fig.3.

L245 I think its vice versa? First the meltwater, then 850 years later the temperature drop?

Thank you for the request for clarification. The final paragraph of this section discusses the relationship of the abrupt cold event to the following freshening event (FE_16.1k). Similarly to above we believe that by providing a more straight forward nomenclature, the text should become more clear.

L276 The "extended discussion" only discusses two other records from Iberia and the Mediterranean, and no ice core records.

We agree that "extended discussion" is misleading here and will remove the annotation in brackets. As discussed in our reply to main comment 3, we will be adjusting which records are compared in the main text vs the extended discussion/supplementary, to compare our record to a larger body of records that have independent age control and have been previously interpreted as (indirect) AMOC strength indicator.

L278 Before jumping to the EASM I would have expected a more comprehensive comparison to other records that are more closer to the North Atlantic realm. Is there a We specific reason why this is missing?

We appreciate the suggestion to clarify the justification for the records included as comparison in Figure 3 and Figure 4.  e will explicitly state our criteria at the onset of the discussion. We use Figure 3 to illustrate the  North Atlantic records relevant to discussion of the Freshening signal.  This figure includes results from sediment cores in the North Atlantic region.  These provide an important context although lacking precise absolute chronology. The figure also presents  Ice sheet model Glac-1D, which is constrained by observations of the Northern Hemisphere ice sheet and provides relevant context  We believe this figure compiles the essential North Atlantic records relevant to interpreting the freshening. As discussed in response to a previous comment, we will provide an alternate set of maps focusing on ice sheet mass loss.

We clarify that the focus of Figure 4 and the resulting discussion, is a focus on the chronology and phasing and therefore includes the relevant records which feature high temporal resolution and an independent absolute age model, comparing regions where the global consequences of AMOC decline have been described. As described in our response to main comment 3, we focus comparison on records from locations where a clear link with

AMOC has been previously described through models, theory, or data, including during AMOC weakening : the cooling in the North Atlantic and western Europe (NGRIP ice core already shown, NALPS Speleothem (Li et al., 2021; Luetscher et al., 2015), the southward ITCZ migration in South America (Brazil Speleothem Paixao Cave (Strikis 2015), the East Asian monsoon response (already Hulu Cave shown in Figure 4), and the atmospheric CO2 rise hypothesized to respond to the Southern Ocean warming during AMOC weakening via a thermal bipolar see saw. Thus Figure 4 will compare:

- Greenland Ice Core (NGRIP)

- WDC CO2

- EASM: Speleothem Hulu YT

- Brazil Speleothem Paixao Cave (Strikis 2015):

- NALPS Speleothem (Li et al., 2021; Luetscher et al., 2015)

,Because changes in ocean circulation hamper assumption about reservoir ages and thus make sediment cores considerably more age uncertain, we are considering them as not suitable for the changepoint analysis in Figure 4.

In contrast, we will mention in the main text, but keep In the supplementrary other available records from N Spain, namely OST2 from Ostolo Cave (Bernal-Wormull et al.,2021) and also Maat from Meravelles Cave. Due to the respective cave systems, d18O in those records is likely more affected by changes in temperature and moisture availability. Because of this added layer of complexity, we considered having these records in the main text would not help our story, thus have put them to the supplementary. We will state this more explicitely in our main text and refer to the supplementary figure.

L290 Any suggestions which "atmospheric patterns"? A more comprehensive discussion could elucidate this possibly?

We will clarify that this sentence continues the discussion of the Westerly Jet from the previous sentences.

L292 This is also another reason why records from that region would be worth to compare with!

We agree that the Cave Without Name record (Feng et al) which records freshwater addition to the Gulf of Mexico, would be interesting to discuss. Unfortunately, this record has a nearly 2 ky hiatus during H1 and would not be included in Fig. 4, but we can use it to provide additional context in the dicussion of Figure 3.

L299 It would be interesting to see if the American records are better to compare with? Studies have suggested a close link to AMOC and NA temperatures - your record could provide the possibility to test this (compare eg Travis Taylor et al., 2025,Warken et al., 2020...)

We will clarify the rationale for which records are included and not in the comparison. We have added to the comparison the record from Brasil (Strikis et al 2015). In other regions of North America and the Caribbean, there are multiple and complex factors contributing to the d18O variability and we consider it beyond the scope of this paper to review and re evaluate the diverse factors encoded in global stalagmite d18O records. Therefore, we hope that our new record will provide an important North Atlantic reference for future studies to test how atmospheric processes in other regions may be affected by AMOC, but such a discussion is beyond the scope of this paper.

As we highlight in our response to main comment 3, we focus on records spanning the full LGM-14 ka period; The work by Travis-Taylor only spans a short section of our record, so is not considered for comparison here.

L301 what is the uncertainty of the ice record here?

The ice core record is based on the WD2014 chronology (Sigl et al., 2016). The age uncertainty in this time interval is better than 1% of the age, thus smaller than 200 years. We will add this information to the sentence in line 301.

L402 Heading does not fit to content, its only one record also from Iberia discussed

We will change the heading to A5 Comparison with Other Iberian Speleothem Records

**References**

Affolter, S., Häuselmann, A., Fleitmann, D., Edwards, R. L., Cheng, H., & Leuenberger, M. (2019). Central Europe temperature constrained by speleothem fluid inclusion water isotopes over the past 14,000 years. Science advances, 5(6), eaav3809.

Cheng, H., Zhang, H., Spötl, C., Baker, J., Sinha, A., Li, H., ... & Edwards, R. L. (2020). Timing and structure of the Younger Dryas event and its underlying climate dynamics. Proceedings of the National Academy of Sciences, 117(38), 23 408-23 417.

Li, H., Spötl, C., & Cheng, H. (2021). A high-resolution speleothem proxy record of the Late Glacial in the European Alps: extending the NALPS19 record until the beginning of the Holocene. Journal of Quaternary Science, 36(1), 29-39.

Luetscher, M., Boch, R., Sodemann, H., Spötl, C., Cheng, H., Edwards, R. L., ... & Müller, W. (2015). North Atlantic storm track changes during the Last Glacial Maximum recorded by Alpine speleothems. Nature Communications, 6(1), 6344.

Stríkis, N. M., et al. (2015), Timing and structure of Mega-SACZ events during Heinrich Stadial 1, Geophys. Res. Lett., 42, 5477–5484, doi:10.1002/2015GL064 048.

Travis-Taylor, L., Medina-Elizalde, M., Pajón, J.M. et al. Hydroclimate variability in the northern Caribbean during the last deglaciation was modulated by large-scale atmospheric circulation and climate events. Commun Earth Environ 6, 498 (2025). https://doi.org/10.1038/s43 247-025-02 465-0

Warken, S.F., Vieten, R., Winter, A., Spötl, C., Miller, T.E., Jochum, K.P., Schröder-Ritzrau, A., Mangini, A. and Scholz, D. (2020), Persistent Link Between Caribbean Precipitation and

Atlantic Ocean Circulation During the Last Glacial Revealed by a Speleothem Record From Puerto Rico. Paleoceanography and Paleoclimatology, 35: e2020PA003 944. https://doi.org/10.1029/2020PA003 944